# FastCar: Cache Attentive Replay for Fast Auto-Regressive Video Generation on the Edge

Xuan Shen[1*]  Weize Ma[2*]  Yufa Zhou[3]  Enhao Tang[2]  Yanyue Xie[1]  Zhengang Li[4]  Yifan Gong[4]
Quanyi Wang[5]  Henghui Ding[6]  Yiwei Wang[7]  Pu Zhao[1†]  Jun Lin[2†]  Jiuxiang Gu[4†]
[1]Northeastern University  [2]Nanjing University  [3]Duke University  [4]Adobe  [5]NUIST
[6]Fudan University  [7]University of California, Merced
shen.xu@northeastern.edu  weizema@smail.nju.edu.cn
jlin@nju.edu.cn  zhao.pu@northeastern.edu  jigu@adobe.com

## Abstract

Auto-regressive (AR) models, initially successful in language generation, have recently shown promise in visual generation tasks due to their superior sampling efficiency. Unlike image generation, video generation requires a substantially larger number of tokens to produce coherent temporal frames, resulting in significant overhead during decoding. We first make specific key observations: (i) MLP modules in the decode phase dominate the inference latency, and (ii) there exists high temporal redundancy in MLP outputs of adjacent frames. With the insights, we propose **FastCar** to accelerate the decode phase for the AR video generation by exploring the temporal redundancy. The Temporal Attention Score (TAS) is proposed to determine whether to apply the replay strategy (*i.e.*, reusing cached MLP outputs from the previous frame to reduce redundant computations) with detailed theoretical analysis and justification. Furthermore, we develop a hardware accelerator on FPGA with Dynamic Resource Scheduling based on TAS to enable better resource utilization and faster inference. Experimental results demonstrate the effectiveness of our method, which outperforms traditional sparse attention approaches with more than $2.1\times$ decoding speedup and higher energy efficiency on the edge. Furthermore, by combining FastCar and sparse attention, FastCar can boost the performance of sparse attention with alleviated drifting, demonstrating our unique advantages for high-resolution and long-duration video generation. Code: https://github.com/shawnricecake/fast-car

## 1 Introduction

Recently, there has been growing interest in extending the Auto-Regressive (AR) framework of Large Language Models (LLMs) (Radford et al., 2019; Touvron et al., 2023; Grattafiori et al., 2024; Zhao et al., 2024a) to visual generation tasks (Sun et al., 2024; Wang et al., 2024a; Han et al., 2024; Tian et al., 2024; Weng et al., 2023; Deng et al., 2024; Jiao et al., 2025; Xie et al., 2024; Sun et al., 2024; Luo et al., 2024; Kondratyuk et al., 2023; Wang et al., 2024b). The works (Sun et al., 2024; Tian et al., 2024; Sun et al., 2024; Han et al., 2024; Wang et al., 2024b) convert images into tokens and apply AR models to generate image tokens with next-token prediction. The generation quality is surprisingly strong, often surpassing diffusion-based techniques in perceptual fidelity and semantic coherence.

As video becomes a dominant medium across entertainment, communication, *etc.*, synthesizing coherent high-quality videos from minimal inputs presents a compelling research challenge (Xiong et al., 2024; Xing et al., 2024; Li et al., 2024a; Melnik et al., 2024). Prior works (Lin et al., 2024a; Zheng et al., 2024; Peng et al., 2025; Hong et al., 2022; Yang et al., 2024; Kong et al., 2024) leverage Diffusion Transformers (DiT) (Peebles & Xie, 2022) to develop video generation models with superior generation performance, at the cost of substantial computations and massive memory demands (He et al., 2025; Jin et al., 2024; Xu et al., 2025; Lu et al., 2025; He et al., 2024; Li et al., 2025b). These characteristics limit their practical applications and deployments for resource-

---

*Equal Contribution
†Corresponding Author

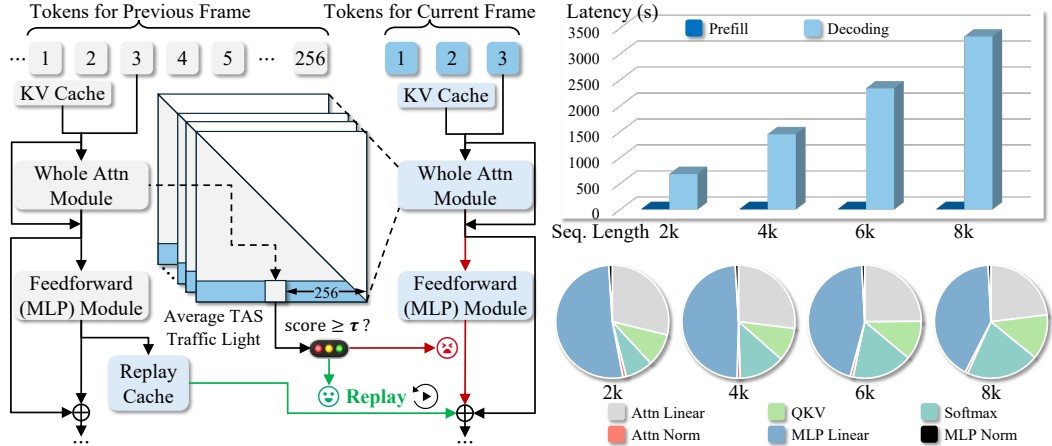

Figure 1: **Left**: FastCar framework. We replay the cache from the previous frame to skip the computations for MLP in decoding. Replay is triggered when the average TAS exceeds a predefined threshold $\tau$. **Right Top**: Latency cost of both prefill and decode phases for different sequence lengths. **Right Bottom**: Detailed latency cost of the decode phase for different sequence lengths.

constrained environments (Liu et al., 2025; Zhu et al., 2023; Li et al., 2023b; Shen et al., 2025e; Yang et al., 2023a; Zhan et al., 2024b; Shen et al., 2025b; Yang et al., 2025) such as mobile devices or Field-Programmable Gate Array (FPGA) with tight constraints for energy efficiency, memory, *etc.*

Motivated by the scalability and fast decoding capabilities of AR transformer-based frameworks in generative tasks, an increasing number of works (Wu et al., 2024; Deng et al., 2024; Weng et al., 2023; Xie et al., 2024; Kondratyuk et al., 2023) have adopted AR frameworks for video generation tasks. To further improve its efficiency, model compression strategies (such as model pruning and quantization (Zhang et al., 2022; Lin et al., 2024b; Ma et al., 2023; Shen et al., 2024a; 2025c; Li et al., 2024b; Zhao et al., 2024b; Zhan et al., 2024c;a; Xiao et al., 2023a; Liang et al., 2025)) and spatial redundancy optimizations (such as sparse attention (Xiao et al., 2023b; Rehg, 2024; Hooper et al., 2024; Liu et al., 2024; Ge et al., 2024; Li et al., 2024c; Shen et al., 2025a; Li et al., 2025a) and efficient sampling methods (Spector & Re, 2023; Yang et al., 2023b; Miao et al., 2023; Ning et al., 2024; Teng et al., 2024; Shen et al., 2025b; He et al., 2025; Kong et al., 2025; Shen et al., 2023; 2024b; 2025d)) are investigated. However, the inherent temporal redundancy specifically introduced by videos with multiple sequential frames remains largely unexplored in AR video generation.

**Specific Deep Insights.** To explore the redundancy for superior efficiency, we first perform a detailed latency profiling and a similarity analysis between different frames. As shown in the right of Figure 1, we identify that the MLP modules (rather than the attention modules) in the decode phase dominate the inference latency. Meanwhile, according to Figure 2, the outputs of adjacent frames for the same MLP module exhibit relatively high resemblance/similarity, indicating high temporal redundancy.

**Framework with Theoretical Justification.** Based on the deep insights specific for AR video generation, we propose **FastCar** for efficiency optimization. The *Temporal Attention Score (TAS)* is proposed to determine whether to skip the computations of the MLP modules (Figure 1). If skipped, the cached outputs from the previous frame are directly reused as current outputs (similar to video replay) due to their high similarity. Skipping computation-intensive MLP modules leads to substantial accelerations. We further provide a detailed theoretical analysis to formally characterize how our TAS controls the output differences across adjacent frames, thereby justifying the design of FastCar.

**Hardware Accelerator.** A flexible and efficiency-oriented hardware accelerator is further developed to support kernel fusion and custom instruction programmability, thus allowing direct reuse of cached outputs and enabling conditional execution of MLP modules. Specifically, to handle varying workload sparsity, we propose *Dynamic Resource Scheduling (DRS)*, which leverages attentivity to dynamically allocate computational resources. DRS, integrated into lightweight control logic, helps alleviate bandwidth pressure and improves overall resource efficiency, thereby enabling faster inference.

**Comprehensive Experiments.** Experimental results show that FastCar not only surpasses sparse attention (SA) methods with better generation quality, but also achieves faster decoding with improved energy efficiency on FPGA. Additionally, FastCar complements SA approaches by mitigating their

drifting issues. By combining FastCar and SA, FastCar significantly boosts the generation quality of SA with faster inference and better long-range temporal coherence.

Our contributions are summarized as follows,

**1.** We perform the latency profiling and similarity analysis between different frames to explore the temporal redundancy in MLP modules. We then propose FastCar framework to accelerate AR video generation by replaying MLP modules using cached outputs from the previous frame.

**2.** Our theoretical analysis demonstrates that the similarity of MLP outputs across adjacent frames correlates with the attentivity, and this correlation is consistent across various model depths, thus justifying the design of FastCar with TAS (*i.e.,* the attentivity) to guide replay decisions.

**3.** We develop an efficiency-oriented hardware accelerator with DRS, enabling dynamic allocation of computational resources to enhance resource utilization and accelerate inference on FPGA.

**4.** Experimental results show that FastCar outperforms SA methods in generation quality by alleviating drifting issues of SA, and achieves more than $2.1\times$ speedup, thereby enhancing scalability and efficiency for high-resolution and long-duration AR video generation.

## 2    RELATED WORK

**Auto-Regressive Visual Generation.** Prior works (Sun et al., 2024; Wang et al., 2024a; Han et al., 2024; Tian et al., 2024; Jiao et al., 2025; Xie et al., 2024; Sun et al., 2024; Luo et al., 2024) apply the AR framework for image generation, demonstrating its potential to outperform diffusion-based models. In particular, VAR (Tian et al., 2024) introduces next-scale prediction to progressively generate token sequences across multiple resolutions, demonstrating the effectiveness of AR methods with enhanced image quality. Inspired by this, several works (Wu et al., 2024; Deng et al., 2024; Weng et al., 2023; Kondratyuk et al., 2023) adopt the AR framework for video generation models.

**Efficient Techniques for Auto-Regressive Visual Generation.** AR image generation models (Sun et al., 2024; Wang et al., 2024a), typically require $n^2$ sequential forward passes to generate an image represented by $n \times n$ tokens, resulting in significant inefficiency, which is further exacerbated when extending to video generation (Wu et al., 2024) with multiple image frames. Some works (He et al., 2024; Teng et al., 2024) accelerate the sampling process at decode phase, utilizing contextual cues from neighboring tokens to reduce redundant computations.

## 3    DEEP INSIGHTS FOR AUTO-REGRESSIVE VIDEO GENERATION

To effectively accelerate AR video generation, we first perform detailed profiling for the inference latency and computations of VILA-U (Wu et al., 2024). We then provide the following specific deep insights: (i) The decode phase takes significantly longer than the prefill phase. (ii) The MLP modules dominate the latency of the decode phase. (iii) The outputs of an MLP module exhibit great similarity to those of its previous frame. Next we demonstrate our detailed observations and analysis.

**Prefill Phase *v.s.* Decode Phase.** We compare the latency of the prefill phase and decode phase under different input sequence lengths from 2k to 8k, during AR video generation. As shown in the right top of Figure 1, the decode phase takes significantly longer than the prefill phase under various input lengths, as it needs to generate a large number of visual tokens for videos with multiple frames.

**Attention Modules *v.s.* MLP Modules.** We further explore detailed latency profiling for different decoding modules. The bottom right of Figure 1 shows that under varying input sequence lengths, MLP modules consistently dominate the overall latency. This observation underscores the distinct computational characteristics of AR generation compared with diffusion-based methods. Specifically, in diffusion transformers, all visual tokens are processed simultaneously through iterative denoising, with attention modules as the primary computational bottleneck. In contrast, AR models generate tokens sequentially, where attention modules only contribute marginally to the overall latency. As a result, efficiency-oriented techniques designed for attention modules are less effective in AR.

**Spatial Redundancy *v.s.* Temporal Redundancy.** Spatial Redundancy is commonly explored in image generation with SA mechanisms (to reduce computations) or efficient sampling (to generate fewer tokens). In contrast, temporal redundancy in video generation remains largely overlooked, as

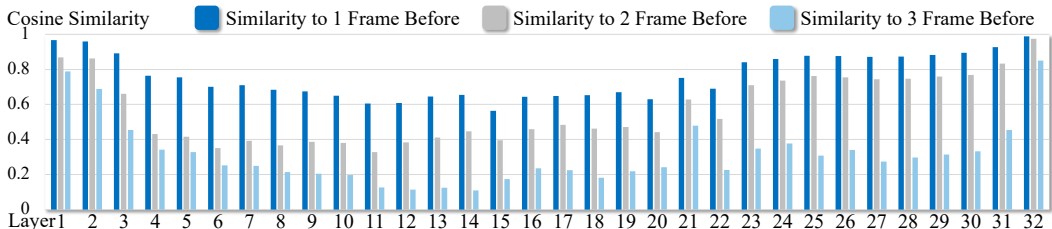

Figure 2: Cosine similarity for MLP outputs between neighboring frames for all 32 MLP modules.

prior works focus on image generation. To explore temporal redundancy, we present cosine similarity between outputs of an MLP module and those of its neighboring frames in Figure 2. The MLP outputs exhibit high similarity with their most recent frame, demonstrating high temporal redundancy.

**Motivation.** Based on the observation that MLP modules dominate the overall latency, we mainly optimize the computations of MLP modules for acceleration. Temporal redundancy with high similarities between the MLP outputs of neighboring frames further motivate us to cache the corresponding hidden states from the current time step for reuse in its next time step, thus avoiding its computations.

## 4 FASTCAR FRAMEWORK DESIGN

We demonstrate our FastCar framework in this section. In general, when the proposed *Temporal Attention Score (TAS)* indicates high similarity, we then directly reuse the cached outputs from its previous frame as the outputs of the current frame, thus skipping the MLP computations.

We first provide specific definitions for AR video generation. Then we demonstrate FastCar in great details. Theoretical analysis are further provided for the rationality and justification of FastCar.

### 4.1 AUTO-REGRESSIVE VIDEO GENERATION

We model a video $\mathcal{V}$ as a temporal sequence of $T$ frames with $N$ visual tokens in each frame. With $\mathcal{V}_{\text{vis}}$ denoting a finite vocabulary of visual tokens, it can be formulated as follows,

$$\mathcal{V} = \{z_{t,i} \mid t = 1, \ldots, T; \ i = 1, \ldots, N\}, \quad z_{t,i} \in \mathcal{V}_{\text{vis}}, \tag{1}$$

Flattening the temporal–spatial grid yields a sequence of length $n = T \cdot N$. We denote the flattened token index as $j = (t, i) := (t-1)N + i$, where $t$ denotes the frame index and $i$ denotes the index of the spatial position. Since frames are consecutively ordered, it satisfies: $(t-1, i) = (t, i) - N$.

For transformer layers, we define the hidden states: $X \in \mathbb{R}^{n \times d}$, where $d$ denotes the hidden size. We use a batch size of $B = 1$ for simplicity, with all results readily extendable to $B > 1$ through broadcasting. The objective of AR video generation is to model the joint distribution:

$$P(\mathcal{V}) = \prod_{j=1}^{n} P(z_j \mid z_{<j}). \tag{2}$$

### 4.2 KEY MODULES

We now formalize the key modules in AR video generation and our FastCar framework. The model has multiple blocks, with an attention module and an MLP module for each block, as defined below.

**Definition 4.1** (Attention Module). *Given hidden states $X \in \mathbb{R}^{n \times d}$, attention output is computed as:*

$$\mathsf{Attn}(X) = \mathsf{Softmax}\left(\frac{QK^{\top}}{\sqrt{d}}\right) V \in \mathbb{R}^{n \times d}, \textit{with } Q = XW_Q, \ K = XW_K, \ V = XW_V, \tag{3}$$

*where $W_Q, W_K, W_V \in \mathbb{R}^{d \times d}$ are the query, key, and value projection matrices.*

**Definition 4.2** (MLP Module). *Given input hidden states $X \in \mathbb{R}^{n \times d}$, the MLP module is defined as:*

$$\mathsf{MLP}(X) = (\mathsf{act}(XW_G) \circ (XW_U)) W_D \in \mathbb{R}^{n \times d}, \tag{4}$$

*where $\mathsf{act}(\cdot)$ is a non-linear activation function (e.g., SiLU), $\circ$ denotes element-wise multiplication, and $W_G, W_U \in \mathbb{R}^{d \times d_{\text{ff}}}$, $W_D \in \mathbb{R}^{d_{\text{ff}} \times d}$ are learned parameters with the intermediate size of $d_{\text{ff}}$.*

Next we define TAS to measure the token similarity of adjacent frames and guide replay decisions.

**Definition 4.3** (Temporal Attention Score). *The* temporal attention score *at spatial position $i$ and $t$-th frame is defined as the scaled dot-product between the query vector $q_j$ of token $j = (t, i)$ and the key vector $k_{j^-}$ of its aligned token $j^- = (t-1, i)$:*

$$s_{t,i} = \frac{\langle q_j, k_{j^-} \rangle}{\sqrt{d}} \in \mathbb{R}. \tag{5}$$

In our FastCar framework, due to causal decoding, TAS is obtained directly from the attention module preceding the MLP, incurring no additional computational cost.

## 4.3 CACHE ATTENTIVE REPLAY FOR FAST GENERATION (FASTCAR)

In FastCar, with TAS, we enable attentive replay in MLP modules by manually setting a threshold $\tau$ to filter tokens of adjacent frames with higher attentivity. When TAS is larger than $\tau$, which indicates significant temporal similarity, we skip MLP computations by reusing the outputs of its previous frame at the same spatial location, as illustrated in the left of Figure 1.

Specifically, for each transformer block, at frame $t - 1$, for each spatial token index $i$, we cache the MLP output as follows:

$$Y_{(t-1,i),:} = \mathsf{MLP}\left(\mathsf{Attn}(X) + X\right)_{(t-1,i),:}. \tag{6}$$

At frame $t$, we evaluate the set of TAS $\{s_{t,i}^{(m)}\}_{m=1}^{h}$ between token $(t, i)$ and its aligned token $(t-1, i)$ across all $h$ attention heads (Definition 4.3), and compute the mean score:

$$\bar{s}_{t,i} = \frac{1}{h} \sum_{m=1}^{h} s_{t,i}^{(m)}. \tag{7}$$

When the mean score exceeds a predefined threshold $\tau$, *i.e.*, $\bar{s}_{t,i} \geq \tau$, we then skip the following MLP computations of this specific token $(t, i)$ and reuse the cached output for the replay in the block:

$$Y_{(t,i),:} = \begin{cases} Y_{(t-1,i),:}, & \text{if } \bar{s}_{t,i} \geq \tau \\ \mathsf{MLP}\left(\mathsf{Attn}(X) + X\right)_{(t,i),:}, & \text{Otherwise} \end{cases}. \tag{8}$$

Otherwise, we still perform the normal MLP computations. In the AR model, there are multiple transformer blocks with the same architecture following the same computation pattern. We apply FastCar for each block. This selective replay mechanism reduces MLP computations by leveraging temporal consistency across adjacent frames.

## 4.4 THEORETICAL SIMILARITY ANALYSIS

We now formally characterize how TAS controls the output differences across adjacent frames, thereby justifying temporal replay based on TAS.

We first relate temporal attention similarity to the difference in attention outputs.

**Theorem 4.4** (Attention Score Controls Attention Output Difference). *Let $X \in \mathbb{R}^{n \times d}$ be the hidden states, where each row $x_j \in \mathbb{R}^d$ represents the hidden state of token $j$. Let $\mathsf{Attn}(X)$ denote the attention output defined in Definition 4.1. For tokens $j = (t, i)$ and $j^- = (t-1, i)$ aligned at the same spatial position, define the temporal attention score $s_{t,i}$ as in Definition 4.3. Assume that:*

- *(1) The hidden states are bounded: $\|x_j\|_2 \leq M$ for all $j$;*

- *(2) The projection matrices satisfy $\|W_Q\|_2, \|W_K\|_2 \leq \Lambda$;*

- *(3) The query and key vectors are normalized: $\|q_j\|_2 = \|k_{j^-}\|_2 = 1$ for all $j$.*

*Let $\gamma := \|W_Q - W_K\|_2$ denote the projection difference.*

*Then, under the Lipschitz continuity of the attention, there exists a constant $C > 0$ such that:*

$$\|\mathsf{Attn}(X)_{j,:} - \mathsf{Attn}(X)_{j^-,:}\|_2 \leq C \left( \sqrt{1 - s_{t,i}} + \gamma M \right). \tag{9}$$

*Thus, a larger $s_{t,i}$ implies a smaller attention output difference up to a model-dependent offset $\gamma M$.*

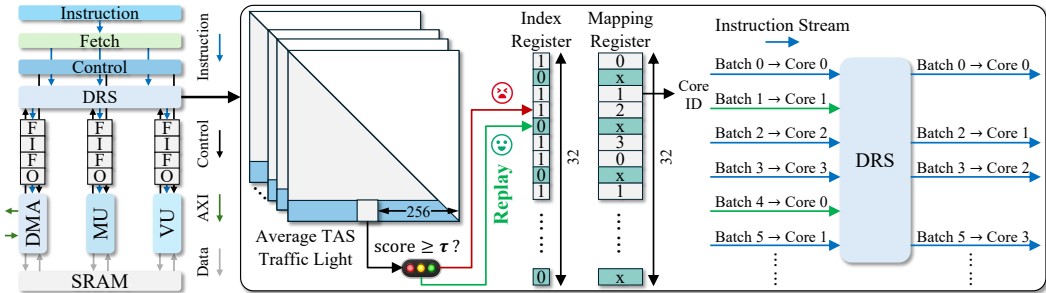

Figure 3: **Left**: The top-level block diagram of our hardware accelerator. **Right**: The DRS diagram.

Transformers often apply normalization techniques (such as LayerNorm or explicit vector normalization) to control query and key magnitudes. Thus assuming $\|q_j\|_2 = \|k_{j^-}\|_2 = 1$ is reasonable and standard for theoretical analysis Shen et al. (2025b;c). We delay the full proof to Appendix C.

Next, we relate input and attention similarity to MLP output similarity.

**Theorem 4.5** (Attention and Input Similarity Implies MLP Output Similarity). *Let* $\mathsf{MLP}(\cdot)$ *denote the MLP module defined in Definition 4.2. Let* $Y_{j,:} = \mathsf{MLP}\left(\mathsf{Attn}(X) + X\right)_{j,:}$ *and* $Y_{j^-,:} = \mathsf{MLP}\left(\mathsf{Attn}(X) + X\right)_{j^-,:}$ *denote the MLP outputs at tokens* $j$ *and* $j^-$.

*Assume that* $\mathsf{MLP}(\cdot)$ *is* $L$*-Lipschitz continuous. Then*

$$\|Y_{j,:} - Y_{j^-,:}\|_2 \le L\left(\|X_{j,:} - X_{j^-,:}\|_2 + \|\mathsf{Attn}(X)_{j,:} - \mathsf{Attn}(X)_{j^-,:}\|_2\right). \tag{10}$$

The proof is demonstrated in Appendix C. Finally, combining the two results, we directly relate TAS to MLP output similarity.

**Theorem 4.6** (Temporal Attention Score Controls MLP Output Similarity). *Let* $X \in \mathbb{R}^{n \times d}$ *be the hidden states, and let* $Y_{j,:}$ *and* $Y_{j^-,:}$ *denote the MLP outputs at tokens* $j$ *and* $j^-$. *Let* $s_{t,i}$ *denote the temporal attention score. Under the assumptions of Theorem 4.4 and assuming the MLP is* $L$*-Lipschitz, there exists a constant* $C > 0$ *such that:*

$$\|Y_{j,:} - Y_{j^-,:}\|_2 \le C\left(\|X_{j,:} - X_{j^-,:}\|_2 + \sqrt{1 - s_{t,i}} + \gamma M\right). \tag{11}$$

The proof is demonstrated in Appendix C. Theorem 4.6 formally establishes that high TAS, combined with input similarity, guarantees small MLP output deviation across adjacent frames. This justifies the use of thresholds on TAS to dynamically skip MLP computations during decoding, enabling efficient temporal replay with controlled quality loss. Moreover, as TAS is computed layer-locally, it offers a stable and depth-independent signal for runtime adaptation (Remark 4.7).

**Remark 4.7.** *The TAS* $s_{t,i}$ *depends only on the local query and key vectors at the current layer and is independent of model depth. It captures instantaneous similarity without accumulating information across layers, making it a stable, efficient, and fine-grained signal for dynamic computation reduction during auto-regressive decoding.*

## 5 HARDWARE DESIGN

We develop a programmable hardware accelerator Shen et al. (2024c); Li et al. (2022); Wu et al. (2022); Zhan et al. (2021); Li et al. (2023a), as shown in the left of Figure 3. Pre-compiled instructions are loaded via the AXI bus with the Fetch module, and dispatched to the corresponding instruction FIFO (First-In-First-Out). The Control module manages the Matrix Unit (MU) and Vector Unit (VU) to perform matrix multiplication and vector computation, while the Direct Memory Access (DMA) module is responsible for loading data from off-chip memory (i.e., DDR or HBM). The instruction FIFO receives control signals from the Control Module to coordinate the computations of each unit. The Dynamic Resource Scheduling (DRS) module is employed to address the computational workload imbalance caused by dynamic replay from the FastCar framework.

Table 1: Main results of our method compared with sparse attention method StreamingLLM Xiao et al. (2023b), where dense attention is retained in the first frame for fair comparison. Local size denotes number of local tokens for sparse attention. Detailed VBench scores are in Appendix A. Latency is tested for whole generation of one video. Power efficiency is computed by GFLOPs/W.

| Method | Replay Ratio | Local Size | PSNR ↑ | SSIM ↑ | LPIPS ↓ | VBench Score | | | TFLOPs ↓ | Latency (s) ↓ | Power Effi. ↑ |
| | | | | | | Total | Quality | Semantic | | | |
|---|---|---|---|---|---|---|---|---|---|---|---|
| Dense | / | / | - | - | - | 74.1% | 76.4% | 65.0% | 31.79 | 689.7 (1×) | 1.47 |
| Sparse Attn. | / | 256 | 18.25 | 51.54 | 33.59 | 72.1% | 74.6% | 62.5% | 30.95 | 670.5 (1.02×) | 1.51 |
| | / | 128 | 13.14 | 33.61 | 54.34 | 60.7% | 61.9% | 55.9% | 30.82 | 666.9 (1.03×) | 1.52 |
| | / | 64 | 13.47 | 33.79 | 53.54 | 62.6% | 63.3% | 60.2% | 30.76 | 666.3 (1.03×) | 1.52 |
| | / | 32 | 13.34 | 33.14 | 53.82 | 61.4% | 61.3% | 62.0% | 30.72 | 663.9 (1.04×) | 1.52 |
| | / | 16 | 13.30 | 32.02 | 53.75 | 64.5% | 64.8% | 63.3% | 30.70 | 662.7 (1.04×) | 1.53 |
| Ours | 10% | / | 18.57 | 53.32 | 27.31 | 73.4% | 75.5% | 65.2% | 30.09 | 629.1 (1.10×) | 1.61 |
| | 20% | / | 17.94 | 51.01 | 27.57 | 73.2% | 75.3% | 65.1% | 28.64 | 556.8 (1.24×) | 1.82 |
| | 30% | / | 17.87 | 50.29 | 28.02 | 72.4% | 74.3% | 64.7% | 27.18 | 525.2 (1.31×) | 1.93 |
| | 40% | / | 17.68 | 50.14 | 28.15 | 71.8% | 73.0% | 67.2% | 25.73 | 487.2 (1.42×) | 2.08 |
| | 50% | / | 17.85 | 50.11 | 28.08 | 71.5% | 72.7% | 66.6% | 24.27 | 475.3 (1.45×) | 2.13 |
| | 60% | / | 17.85 | 50.55 | 28.72 | 71.4% | 72.7% | 66.2% | 22.33 | 451.9 (1.53×) | 2.24 |
| | 70% | / | 17.86 | 50.18 | 28.79 | 71.2% | 72.3% | 66.9% | 20.88 | 415.8 (1.66×) | 2.43 |
| | 80% | / | 17.71 | 49.01 | 29.50 | 71.5% | 73.0% | 65.6% | 19.42 | 390.7 (1.76×) | 2.59 |

## DYNAMIC RESOURCE SCHEDULING (DRS)

The FastCar framework dynamically determines whether to adopt the replay strategy to skip computations for certain MLP modules based on the TAS of different batches. Due to the multi-core design, computations for different batches in dense mode are mapped to distinct cores statically. However, the dynamic FastCar introduces computational workload imbalance across different cores, as the number of MLPs computed for different batches may vary and is difficult to predict during inference. Moreover, we employ static compilation to pre-generate scheduling instructions. Exhaustively enumerating all possible cases would incur an unaffordable large instruction storage overhead.

To address this, we propose the DRS to balance the computational workloads, as shown in the right of Figure 3. After computing TAS, we configure an on-chip computation mapping table. A 32-bit Index Register, which stores the status of each batch (0=replay, 1=compute), is established to determine whether computation should be skipped. 32 Mapping Registers, each with $\log_2(\text{num\_cores})$ bits, determine the target core for executing which batch. We adopt a round-robin assignment strategy to ensure balanced workload distribution among the cores.

When pre-compiled instructions are loaded and the replay mechanism is triggered, the prefetched instructions are forwarded to the DRS for processing. The DRS then performs scheduling decisions by consulting the Index Register to determine whether to discard instructions for replayed batches or dispatch them to the appropriate cores based on the Mapping Registers. Notably, the DRS incurs minimal overhead by completing its dispatch operations in just hundreds to thousands of cycles, which is negligible compared to the thousands of cycles required for actual instruction execution.

# 6 EXPERIMENTAL RESULTS

## 6.1 EXPERIMENTAL SETUP

We adopt the AR video generation model from VILA-U (Wu et al., 2024). Videos are generated with 8 frames in 256×256 resolution, where each frame is decoded by 256 tokens. VILA-U is the only available open-source model at this time (Section 2). We evaluate quality of generated videos with VBench (Huang et al., 2024), and the similarity metrics including PSNR, SSIM, and LPIPS (Zhang et al., 2018). In detail, we compute the average similarity across all frames except the first one, as our method generates the first frame in the same manner as the baseline. We generate videos using prompts from VBench with a batch size of 5, a fixed random seed of 42, and a classifier-free guidance of 3 (Ho & Salimans, 2022) on A100 GPUs. Additionally, there are no available direct baselines in this novel area, and we compare our method against the sparse attention (SA) approach StreamingLLM (Xiao et al., 2023b). We set the sink size by extending the prefill length by 256 to ensure the first frame is preserved throughout video generation for fair comparison.

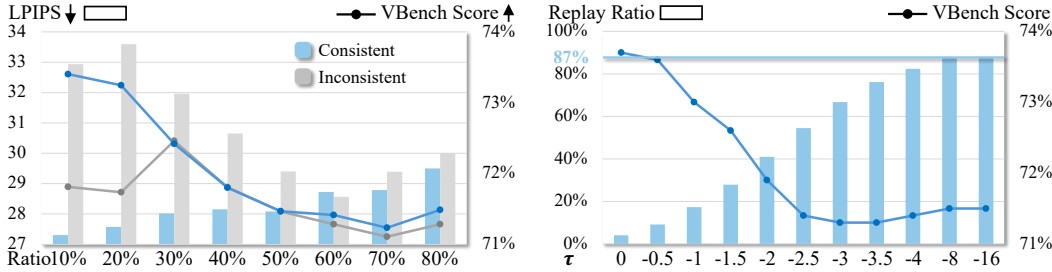

Figure 4: **Left:** Ablation study comparing consistent vs. inconsistent threshold settings with respect to LPIPS and the VBench total score. Full results are provided in Table 5 at Appendix A. **Right:** Ablation study on the effect of the threshold $\tau$ on replay ratio and VBench total score. Full results are reported in Table 6 at Appendix A.

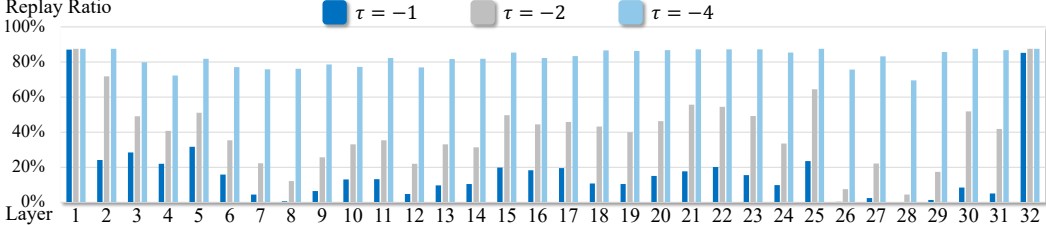

Figure 5: Replay ratio distribution across layers for thresholds $\tau = -1, -2$, and $-4$, respectively.

For hardware implementation, we adopt the Xilinx Alveo U280 FPGA as the target platform with a chiplet design. We implement multiple accelerator cores on the FPGA to ensure physical implementation feasibility. Latency and power are tested using a prefill sequence length of 256.

## 6.2 MAIN RESULTS

We provide the main results with different replay ratios compared with the SA method in Table 1. The detailed VBench scores for all results in different dimensions are included in Appendix A. We make the following observations: (i) The SA methods are not able to accelerate AR video generation models effectively as the MLP modules dominate the computations/latency, and thus optimizing attention modules does not effectively address the bottleneck. (ii) Our method effectively reduces the computations and achieves significant accelerations with better power efficiency. With an 80% replay ratio, our method can reduce 45% computations with a $1.77\times$ acceleration. (iii) Our method better maintains the generation quality than the SA methods. For similarity metrics including PSNR, SSIM, and LPIPS, with gradually increasing sparsity, our generation quality only degrades marginally, which significantly outperforms the SA method with notable degradations. For the VBench scores, we can make similar observations. Meanwhile, our method achieves higher power efficiency compared to the SA method. Also, we provide the visualization with our method in different replay ratios in Appendix B. Visualization shows that video quality is well preserved across different replay ratios.

## 6.3 ABLATION STUDY

**Threshold Distributions.** We conduct an ablation study on threshold distribution by applying either consistent or layer-wise varying (i.e., inconsistent) thresholds across all layers, while maintaining the same overall replay ratio. As shown in the left side of Figure 4, consistent threshold achieves better performance with lower LPIPS and higher VBench score than inconsistent thresholds, which verifies the effectiveness of Remark 4.7. More discussions can be found in Section A.2 of Appendix.

**Threshold Values.** Meanwhile, we ablate the threshold values to evaluate their impact on model performance, as shown in the right side of Figure 4. When $\tau \leq -2.5$, if we continue to decrease $\tau$, the generation quality does not further degrade while higher sparsity with faster inference can be achieved, demonstrating the robustness of FastCar. Additionally, we observe that the AR video generation model achieves the highest replay ratio of 87% when $\tau \approx -8$, indicating that only 13% of the MLP modules are actually required during the generation process.

Table 2: Additional results for the combination of the sparse attention method and our method. More results are included in Table 7 at Appendix A.

| Method | Replay Ratio | Local Size | PSNR ↑ | SSIM ↑ | LPIPS ↓ | VBench Score | | | GFLOPs ↓ | Latency (s)↓ | Power Effi.↑ |
|---|---|---|---|---|---|---|---|---|---|---|---|
| | | | | | | Total | Quality | Semantic | | | |
| Dense | / | / | - | - | - | 74.1% | 76.4% | 65.0% | 31.79 | 689.7 (1×) | 1.47 |
| Ours + Sparse Attn. | 87% | 256 | 17.44 | 47.57 | 31.27 | 71.8% | 73.3% | 65.7% | 17.61 | 354.46 (1.95×) | 2.85 |
| | 87% | 128 | 17.29 | 46.79 | 32.10 | 71.7% | 73.3% | 65.7% | 17.49 | 342.25 (2.02×) | 2.96 |
| | 87% | 64 | 17.27 | 46.70 | 32.09 | 71.6% | 73.1% | 65.5% | 17.43 | 334.57 (2.06×) | 3.02 |
| | 87% | 32 | 17.27 | 46.75 | 31.96 | 71.9% | 73.4% | 65.9% | 17.40 | 327.36 (2.11×) | 3.09 |
| | 87% | 16 | 17.27 | 46.49 | 32.37 | 71.6% | 73.1% | 65.7% | 17.39 | 324.31 (2.13×) | 3.12 |

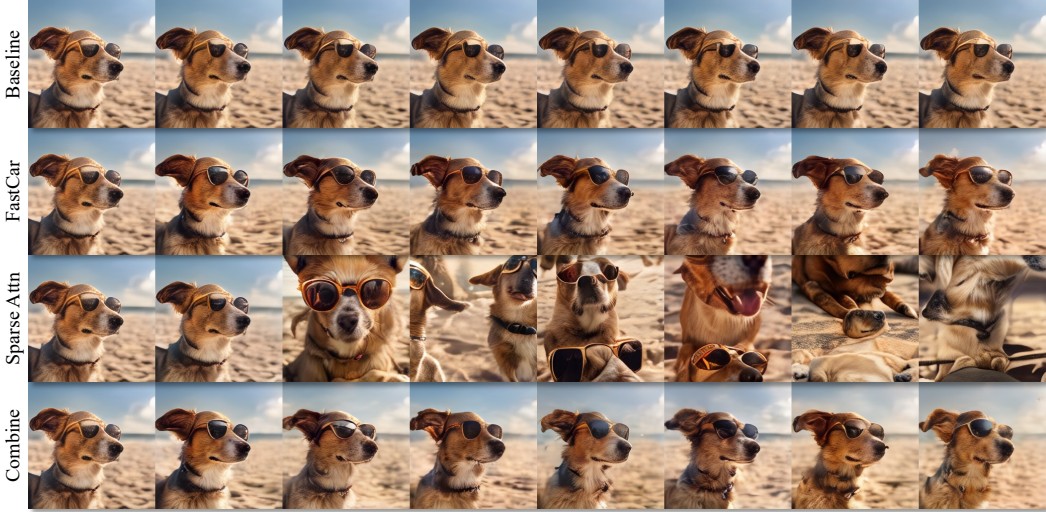

Figure 6: Visualization for the prompt *"A dog wearing sunglasses on the beach."*. The second and third rows are generated with threshold $\tau = -4$ (i.e., 82% replay ratio). The third and fourth rows are generated with a sink size that extends the prompt length by one frame and 64.

## 6.4 ADDITIONAL ANALYSIS

**Replay Ratio Distribution.** We visualize the replay ratio distribution across layers for 3 different thresholds in Figure 5. We observe that the model tends to replay at the shallow and deep layers, while replay is less likely to occur in intermediate layers. This indicates that intermediate layers play a critical role in capturing temporal dynamics and contribute most significantly to generation quality in auto-regressive video models.

**Combination with Sparse Attention.** We further provide additional results achieved by combining SA method and our method in Table 2. The detailed VBench scores for all results in different dimensions are included in Table 7 at Appendix A. The results show that our method can significantly boost the performance of the SA method through a straightforward combination. This validates the effectiveness of our method as a complementary enhancement to existing SA approaches. We further visualize the results of our method, the SA approach, and their combination in Figure 6 to directly illustrate how our method alleviates drifting when integrated into sparse attention.

## 7 CONCLUSION

We propose FastCar, a framework to accelerate AR video generation on the edge. We show that the similarity across adjacent frames correlates with attentivity and is independent with model depths. We then introduce a replay strategy that reuses cached MLP outputs from the previous frame to reduce computation. Furthermore, the DRS design is adopted to improve resource utilization and inference speed on FPGAs. Results show that our method outperforms SA approaches and complements them with more than 2.1× speedup, enabling better scalability for high-resolution, long-duration video generation. In future work, we plan to extend our framework to a broader range of model families.

## REPRODUCIBILITY STATEMENT

The proposed framework is developed on top of auto-regressive video generation models and leverages the reuse of MLP outputs to reduce computation. The theoretical analysis supporting the similarity of MLP outputs across adjacent frames correlates with the attentivity is provided in the paper. All hardware metrics, including latency and power efficiency, are empirically evaluated on real devices. The full codebase and implementation details will be released publicly upon acceptance of the paper.

## LLMS USAGE STATEMENT

We report that LLMs were used only for stylistic editing of the manuscript text. All scientific content, analysis, and conclusions remain the sole work of the authors.

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

APPENDIX

## A  ADDITIONAL RESULTS

### A.1  DETAILED RESULTS FOR VBENCH

We provide the detailed scores of VBench in Table 3 and Table 4. Our method better maintains the generation quality than SA methods. Specifically, on the VBench, when increasing the sparsity, unlike SA method with a significant drop on most of subtasks, our method keeps high scores close to the dense model on most subtasks under various sparsity.

Table 3: Detailed VBench scores.

| Method | Replay Ratio | Local Size | Overall Consistency | Subject Consistency | Background Consistency | Temporal Flickering | Motion Smoothness | Dynamic Degree | Aesthetic Quality | Imaging Quality |
|---|---|---|---|---|---|---|---|---|---|---|
| Dense | / | / | 27.9% | 87.8% | 94.6% | 95.8% | 94.9% | 59.4% | 57.4% | 58.8% |
| Sparse Attn. | / | 16 | 28.0% | 71.9% | 88.0% | 84.5% | 84.8% | 100.0% | 54.2% | 59.1% |
| | / | 32 | 27.9% | 66.8% | 86.2% | 82.5% | 82.3% | 100.0% | 54.2% | 59.0% |
| | / | 64 | 27.7% | 67.4% | 86.3% | 84.6% | 84.4% | 100.0% | 54.0% | 58.0% |
| | / | 128 | 27.5% | 63.7% | 85.1% | 84.4% | 84.3% | 99.7% | 52.9% | 57.2% |
| | / | 256 | 27.8% | 82.7% | 92.8% | 92.5% | 92.4% | 94.4% | 55.9% | 56.8% |
| Ours | 10% | / | 27.7% | 86.2% | 94.0% | 95.9% | 94.6% | 57.8% | 57.0% | 57.5% |
| | 20% | / | 27.9% | 86.7% | 93.7% | 95.5% | 94.7% | 57.5% | 56.9% | 57.4% |
| | 30% | / | 27.9% | 85.7% | 93.4% | 95.1% | 94.4% | 52.8% | 57.0% | 57.1% |
| | 40% | / | 27.9% | 87.1% | 93.7% | 95.0% | 94.9% | 35.6% | 57.2% | 57.5% |
| | 50% | / | 27.9% | 88.2% | 94.1% | 95.1% | 95.0% | 20.6% | 57.7% | 57.8% |
| | 60% | / | 27.9% | 89.3% | 94.2% | 94.9% | 95.2% | 12.8% | 57.9% | 58.0% |
| | 70% | / | 28.1% | 89.4% | 94.1% | 94.4% | 95.0% | 11.4% | 58.2% | 58.3% |
| | 80% | / | 28.2% | 88.7% | 93.8% | 93.7% | 94.4% | 23.3% | 58.2% | 58.5% |

Table 4: Detailed VBench scores.

| Method | Replay Ratio | Local Size | Object Class | Multiple Objects | Human Action | Color | Spatial Relationship | Scene | Appearance Style | Temporal Style |
|---|---|---|---|---|---|---|---|---|---|---|
| Dense | / | / | 76.7% | 30.8% | 74.8% | 82.0% | 37.6% | 42.6% | 24.7% | 25.0% |
| Sparse Attn. | / | 16 | 67.8% | 20.5% | 81.8% | 81.7% | 41.2% | 37.3% | 24.6% | 24.9% |
| | / | 32 | 65.7% | 19.0% | 83.2% | 80.0% | 37.2% | 35.0% | 24.5% | 24.8% |
| | / | 64 | 61.9% | 17.2% | 81.8% | 77.6% | 30.9% | 33.7% | 24.5% | 25.2% |
| | / | 128 | 50.1% | 12.0% | 78.8% | 73.4% | 19.9% | 32.6% | 24.3% | 25.1% |
| | / | 256 | 68.1% | 26.6% | 80.4% | 76.7% | 31.8% | 38.9% | 24.5% | 25.3% |
| Ours | 10% | / | 74.9% | 33.7% | 76.8% | 78.6% | 42.2% | 40.4% | 24.7% | 24.9% |
| | 20% | / | 74.0% | 34.0% | 76.0% | 78.9% | 41.0% | 40.6% | 24.7% | 25.1% |
| | 30% | / | 74.4% | 32.8% | 75.8% | 78.0% | 39.8% | 40.8% | 24.8% | 25.1% |
| | 40% | / | 76.5% | 31.0% | 73.8% | 76.1% | 40.9% | 40.6% | 24.7% | 25.2% |
| | 50% | / | 77.1% | 34.3% | 74.0% | 76.8% | 42.7% | 42.0% | 24.8% | 25.2% |
| | 60% | / | 77.6% | 36.0% | 75.4% | 79.2% | 43.5% | 42.2% | 24.8% | 25.3% |
| | 70% | / | 77.9% | 36.3% | 79.0% | 79.2% | 43.8% | 42.6% | 24.8% | 25.3% |
| | 80% | / | 77.9% | 36.3% | 75.4% | 77.3% | 44.9% | 42.3% | 24.7% | 25.4% |

### A.2  DETAILED ABLATION RESULTS FOR THRESHOLD DISTRIBUTION

We provide full results for the ablation of threshold distribution in Table 5. We observe that consistent threshold achieves better performance with lower LPIPS and higher VBench score than inconsistent thresholds, which verifies the effectiveness of Remark 4.7 discussed in Section 4.4.

**Inconsistent Thresholds.** The experimental results in Figure 4 include both consistent threshold and inconsistent threshold settings. However, we emphasize that the inconsistent threshold setting used in our experiment is not a truly adaptive thresholding strategy. In our setup, the inconsistent thresholds were manually assigned across layers according to the replay ratio distributions as shown in Figure 5. This configuration mimics an uneven thresholding scheme but was designed to maintain the same overall replay ratio as the consistent threshold baseline, allowing for a controlled comparison. Although they are inconsistent, they are still fixed, and not optimal or adaptive.

A genuinely adaptive thresholding mechanism, which dynamically adjusts thresholds based on token importance (e.g., derived from attention score distributions), may potentially lead to better

performance than consistent threshold. Our work focus on exploring the temporal redundancy in MLP modules using a plug-and-play cache replay strategy with consistent threshold simplifying deployments and achieving better performance. We leave the promising direction to explore more-advanced attention-aware adaptive thresholds as our future work.

**Performance of Inconsistent Thresholds.** In Figure 4 (left), we observe that according to the LPIPS score for the inconsistent thresholds, the performance initially improves as the replay ratio increases, and eventually degrades at very high replay levels. We highlight that since inconsistent thresholds are not optimal or adaptive as discussed above, we still recommend to use consistent threshold with better performance. This trend of inconsistent thresholds can be explained by the fact that our proposed method helps reduce temporal drifting in the generated videos, as discussed in Section 6.4.

More specifically, moderate replaying of MLP outputs introduces temporal consistency across frames, which helps suppress frame-to-frame inconsistencies and improves perceptual similarity. This effect is visually evident in Figure 6, where our method exhibits minimal drifting compared to sparse attention-based baselines, which suffer from severe drift artifacts. Furthermore, when our method is combined with sparse attention, it can help mitigate the drifting effects introduced by sparse attention, leading to improved overall visual coherence in the generated videos.

However, as the replay ratio becomes too large, over-replaying leads to excessive reuse of stale information, which degrades generation quality. Thus, there exists an optimal replay ratio where replay enhances consistency without sacrificing content fidelity, explaining the observed trend in LPIPS.

## A.3    Detailed Ablation Results for Threshold Values

We provide full results for the ablation of threshold values in Table 6. When $\tau \leq -2.5$, if we continue to decrease $\tau$, the generation quality does not further degrade while higher sparsity with faster inference is achieved, demonstrating the robustness of FastCar. Additionally, we observe that the AR video generation model achieves the highest replay ratio of 87% when $\tau \approx -8$, indicating that only 13% of the MLP modules are actually required during the generation process.

## A.4    Full Results for Additional Analysis

We provide full results for the combination of the sparse attention method and our method in Table 7. The results show that our method significantly boosts the performance of SA method through the straightforward combination. This validates the effectiveness of our method as a complementary enhancement to existing SA approaches.

Table 5: Full results for the ablation of the threshold distribution.

| Threshold Distribution | Replay Ratio | PSNR ↑ | SSIM ↑ | LPIPS ↓ | VBench Score | | |
|---|---|---|---|---|---|---|---|
| | | | | | Total | Quality | Semantic |
| Consistent | 10% | 18.57 | 53.32 | 27.31 | 73.4% | 75.5% | 65.2% |
| Inconsistent | 10% | 16.73 | 46.63 | 32.94 | 71.8% | 73.7% | 64.3% |
| Consistent | 20% | 17.94 | 51.01 | 27.57 | 73.2% | 75.3% | 65.1% |
| Inconsistent | 20% | 16.30 | 45.05 | 33.60 | 71.7% | 73.3% | 65.4% |
| Consistent | 30% | 17.87 | 50.29 | 28.02 | 72.4% | 74.3% | 64.7% |
| Inconsistent | 30% | 16.67 | 45.39 | 31.96 | 72.5% | 74.0% | 66.5% |
| Consistent | 40% | 17.68 | 50.14 | 28.15 | 71.8% | 73.0% | 67.2% |
| Inconsistent | 40% | 17.34 | 48.61 | 30.65 | 71.8% | 73.6% | 64.5% |
| Consistent | 50% | 17.85 | 50.11 | 28.08 | 71.5% | 72.7% | 66.6% |
| Inconsistent | 50% | 17.65 | 49.94 | 29.40 | 71.5% | 73.0% | 65.4% |
| Consistent | 60% | 17.85 | 50.55 | 28.72 | 71.4% | 72.7% | 66.2% |
| Inconsistent | 60% | 17.79 | 49.55 | 28.56 | 71.3% | 72.5% | 66.5% |
| Consistent | 70% | 17.86 | 50.18 | 28.79 | 71.2% | 72.3% | 66.9% |
| Inconsistent | 70% | 17.68 | 48.84 | 29.39 | 71.1% | 72.4% | 65.9% |
| Consistent | 80% | 17.71 | 49.01 | 29.50 | 71.5% | 73.0% | 65.6% |
| Inconsistent | 80% | 17.59 | 48.06 | 30.01 | 71.3% | 72.5% | 66.3% |

Table 6: Full results for the ablation of the threshold values.

| Threshold Values | Replay Ratio | PSNR ↑ | SSIM ↑ | LPIPS ↓ | VBench Score | | |
|---|---|---|---|---|---|---|---|
| | | | | | Total | Quality | Semantic |
| 0 | 3.96% | 19.71 | 57.66 | 24.14 | 73.7% | 75.7% | 65.8% |
| -0.5 | 9.13% | 18.61 | 53.49 | 27.27 | 73.6% | 75.6% | 65.5% |
| -1 | 17.32% | 17.84 | 50.49 | 29.22 | 73.0% | 75.1% | 64.6% |
| -1.5 | 27.81% | 17.31 | 48.39 | 30.85 | 72.6% | 74.5% | 64.7% |
| -2 | 40.92% | 17.38 | 48.84 | 30.45 | 71.9% | 73.6% | 65.2% |
| -2.5 | 54.55% | 17.76 | 50.30 | 29.05 | 71.4% | 72.7% | 65.8% |
| -3 | 66.78% | 17.87 | 50.42 | 28.69 | 71.3% | 72.5% | 66.3% |
| -3.5 | 76.20% | 17.76 | 49.42 | 29.26 | 71.3% | 72.6% | 66.2% |
| -4 | 82.41% | 17.65 | 48.51 | 29.83 | 71.4% | 72.7% | 66.2% |
| -8 | 87.49% | 17.60 | 48.50 | 30.09 | 71.5% | 72.9% | 66.0% |
| -16 | 87.49% | 17.60 | 48.04 | 30.09 | 71.5% | 72.9% | 66.0% |

Table 7: Full results for the combination of the sparse attention method and our method.

| Method | Threshold Value | Replay Ratio | Local Size | PSNR ↑ | SSIM ↑ | LPIPS ↓ | VBench Score | | | TFLOPs ↓ | Latency (s) ↓ | Power Effi. ↑ |
|---|---|---|---|---|---|---|---|---|---|---|---|---|
| | | | | | | | Total | Quality | Semantic | | | |
| Dense | / | / | / | - | - | - | 74.1% | 76.4% | 65.0% | 31.79 | 689.71 | 1.47 |
| Ours + Sparse Attn. | -1 | 17.72% | 16 | 12.96 | 29.57 | 55.13 | 60.8% | 61.6% | 57.8% | 28.05 | 497.35 | 2.03 |
| | -2 | 46.11% | 16 | 14.38 | 36.14 | 47.50 | 64.7% | 66.5% | 57.5% | 23.69 | 427.75 | 2.36 |
| | -3 | 70.23% | 16 | 16.95 | 45.72 | 34.25 | 70.5% | 72.0% | 64.9% | 19.81 | 356.35 | 2.84 |
| | -4 | 83.85% | 16 | 17.27 | 46.49 | 32.37 | 71.6% | 73.1% | 65.7% | 17.87 | 331.29 | 3.05 |
| Ours + Sparse Attn. | -1 | 17.72% | 32 | 13.25 | 31.58 | 53.77 | 61.2% | 61.9% | 58.2% | 28.07 | 499.79 | 2.02 |
| | -2 | 46.11% | 32 | 14.43 | 36.81 | 47.18 | 65.0% | 66.6% | 59.0% | 23.71 | 430.18 | 2.35 |
| | -3 | 70.23% | 32 | 16.94 | 46.02 | 34.09 | 70.8% | 72.3% | 64.9% | 19.83 | 358.78 | 2.82 |
| | -4 | 83.85% | 32 | 17.27 | 46.75 | 31.96 | 71.9% | 73.4% | 65.9% | 17.89 | 333.73 | 3.03 |
| Ours + Sparse Attn. | -1 | 17.72% | 64 | 13.25 | 31.90 | 54.32 | 60.0% | 60.9% | 56.5% | 28.10 | 504.60 | 2.00 |
| | -2 | 46.11% | 64 | 14.41 | 36.83 | 47.72 | 64.6% | 66.2% | 58.4% | 23.74 | 435.00 | 2.33 |
| | -3 | 70.23% | 64 | 16.88 | 45.87 | 34.60 | 70.4% | 71.8% | 64.7% | 19.86 | 363.60 | 2.78 |
| | -4 | 83.85% | 64 | 17.27 | 46.70 | 32.09 | 71.6% | 73.1% | 65.5% | 17.92 | 338.55 | 2.99 |
| Ours + Sparse Attn. | -1 | 17.72% | 128 | 13.14 | 31.63 | 55.02 | 59.1% | 60.3% | 53.9% | 28.16 | 514.00 | 1.97 |
| | -2 | 46.11% | 128 | 14.44 | 37.20 | 47.80 | 64.2% | 65.9% | 57.6% | 23.80 | 444.40 | 2.28 |
| | -3 | 70.23% | 128 | 16.89 | 46.02 | 34.77 | 70.0% | 71.5% | 63.9% | 19.92 | 373.00 | 2.71 |
| | -4 | 83.85% | 128 | 17.29 | 46.79 | 32.10 | 71.7% | 73.3% | 65.7% | 17.98 | 347.95 | 2.91 |
| Ours + Sparse Attn. | -1 | 17.72% | 256 | 15.22 | 40.77 | 44.61 | 67.7% | 70.0% | 58.7% | 28.28 | 531.87 | 1.90 |
| | -2 | 46.11% | 256 | 15.78 | 43.05 | 40.25 | 68.9% | 71.1% | 60.1% | 23.92 | 462.27 | 2.19 |
| | -3 | 70.23% | 256 | 17.21 | 47.63 | 32.94 | 70.7% | 72.2% | 64.8% | 20.04 | 390.87 | 2.59 |
| | -4 | 83.85% | 256 | 17.44 | 47.57 | 31.27 | 71.8% | 73.3% | 65.7% | 18.10 | 365.82 | 2.76 |

# B ADDITIONAL VISUALIZATION

We visualize the results of our method under different replay ratios. Our method generates high quality videos.

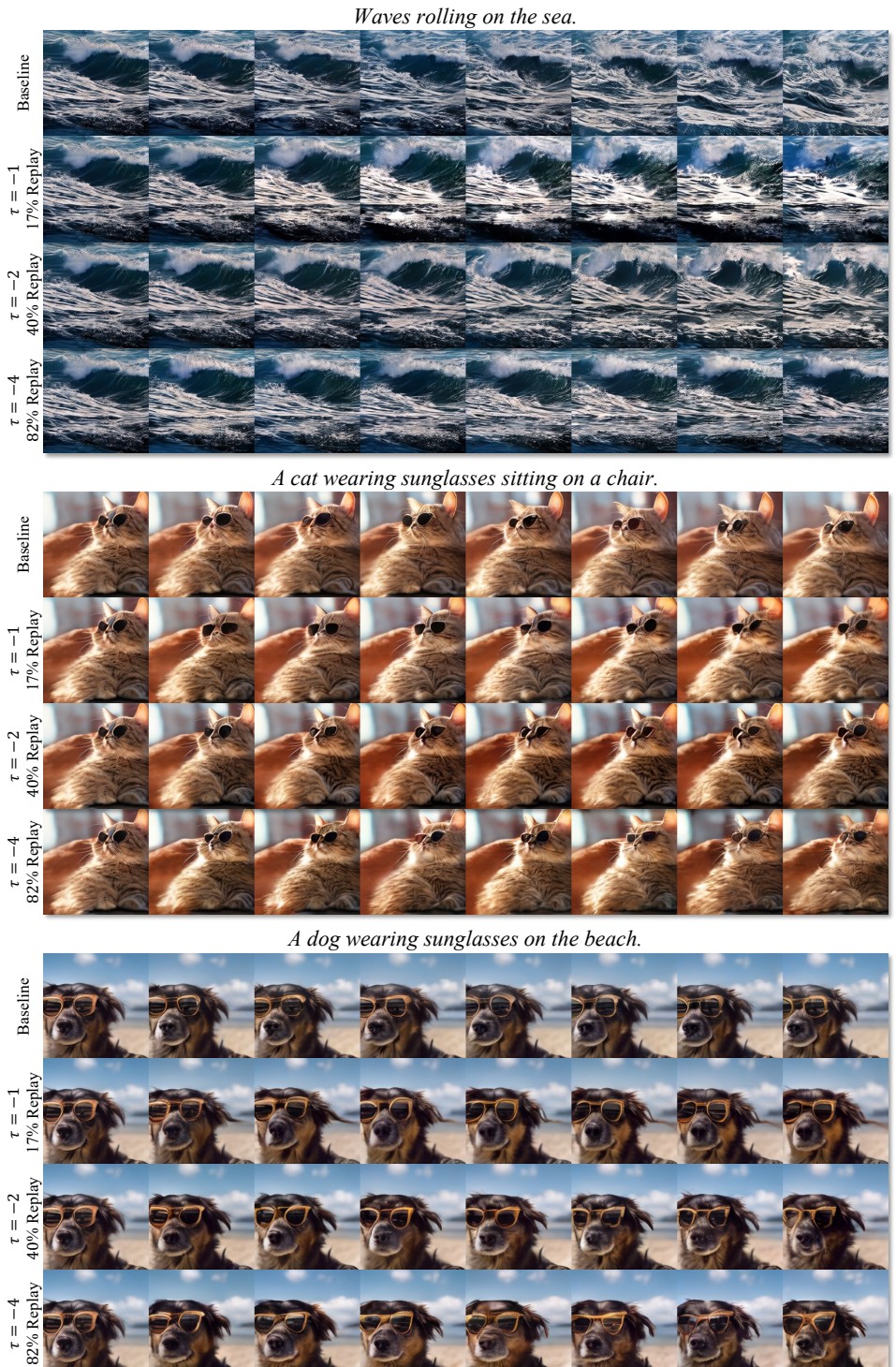

Figure 7: Additional visualization with threshold $\tau = -1, -2, -4$.

## C  DETAILED PROOFS

### C.1  PROOF OF THEOREM 4.4

*Proof of Theorem 4.4.* **Step 1 (Score exactly matches cosine similarity).** By Definition 4.3, $s_{t,i} = \langle q_j, k_{j-} \rangle / \sqrt{d}$, where $q_j = x_j W_Q$ and $k_{j-} = x_{j-} W_K$. Under Assumption (3), $\|q_j\|_2 = \|k_{j-}\|_2 = 1$, so $s_{t,i}$ (up to $\sqrt{d}$ scaling) equals the cosine similarity:

$$\cos\theta(q_j, k_{j-}) = \langle q_j, k_{j-} \rangle.$$

Thus, by the Law of Cosines for unit vectors,

$$\|q_j - k_{j-}\|_2^2 = 2(1 - s_{t,i}).$$

**Step 2 (Logit gap from query gap).** The attention logits satisfy

$$\ell_j = q_j K^\top, \quad \ell_{j-} = q_{j-} K^\top,$$

thus

$$\|\ell_j - \ell_{j-}\|_2 = \|(q_j - q_{j-})K^\top\|_2$$
$$\leq \|K\|_2 \|q_j - q_{j-}\|_2,$$

where $K = X W_K$ is the stacked key matrix. Since $K = X W_K$, we have

$$\|K\|_2 \leq \|X\|_2 \|W_K\|_2 \leq \sqrt{n} M \Lambda,$$

where $\|X\|_2 \leq \sqrt{n}M$ since each $\|x_j\|_2 \leq M$.

**Step 3 (Attention output is Lipschitz).** Since softmax and value projection are Lipschitz continuous (see Shen et al. (2025b)), there exists $L_{\text{attn}} > 0$ such that

$$\|\mathsf{Attn}(X)_{j,:} - \mathsf{Attn}(X)_{j-,:}\|_2 \leq L_{\text{attn}} \|\ell_j - \ell_{j-}\|_2 \leq C_1 \|q_j - q_{j-}\|_2,$$

where $C_1 = L_{\text{attn}} \sqrt{n} M \Lambda$.

**Step 4 (Bounding query–key difference).** Since

$$q_{j-} = x_{j-} W_Q, \quad k_{j-} = x_{j-} W_K,$$

it follows that

$$\|k_{j-} - q_{j-}\|_2 = \|x_{j-}(W_K - W_Q)\|_2 \leq \gamma \|x_{j-}\|_2 \leq \gamma M.$$

By triangle inequality,

$$\|q_j - q_{j-}\|_2 \leq \|q_j - k_{j-}\|_2 + \|k_{j-} - q_{j-}\|_2 \leq \sqrt{2(1 - s_{t,i})} + \gamma M.$$

**Step 5 (Final bound).** Thus,

$$\|\mathsf{Attn}(X)_{j,:} - \mathsf{Attn}(X)_{j-,:}\|_2 \leq C_1\left(\sqrt{2(1 - s_{t,i})} + \gamma M\right)$$
$$\leq C\left(\sqrt{1 - s_{t,i}} + \gamma M\right),$$

after absorbing constants into $C > 0$. This completes the proof. $\square$

### C.2  PROOF OF THEOREM 4.5

*Proof of Theorem 4.5.* Define

$$Z_j = \mathsf{Attn}(X)_{j,:} + X_{j,:}, \quad Z_{j-} = \mathsf{Attn}(X)_{j-,:} + X_{j-,:}.$$

Then

$$Y_{j,:} = \mathsf{MLP}(Z_j), \quad Y_{j-,:} = \mathsf{MLP}(Z_{j-}).$$

By Lipschitz continuity of MLP,

$$\|Y_{j,:} - Y_{j-,:}\|_2 \leq L \|Z_j - Z_{j-}\|_2.$$

Expanding $Z_j - Z_{j-}$ and applying triangle inequality,

$$\|Z_j - Z_{j-}\|_2 \leq \|\mathsf{Attn}(X)_{j,:} - \mathsf{Attn}(X)_{j-,:}\|_2 + \|X_{j,:} - X_{j-,:}\|_2.$$

The claim follows. $\square$

## C.3 Proof of Theorem 4.6

*Proof of Theorem 4.6.* By Theorem 4.5,

$$\|Y_{j,:} - Y_{j^-,:}\|_2 \leq L\left(\|X_{j,:} - X_{j^-,:}\|_2 + \|\text{Attn}(X)_{j,:} - \text{Attn}(X)_{j^-,:}\|_2\right).$$

By Theorem 4.4,

$$\|\text{Attn}(X)_{j,:} - \text{Attn}(X)_{j^-,:}\|_2 \leq C'\left(\sqrt{1 - s_{t,i}} + \gamma M\right).$$

Substituting gives

$$\|Y_{j,:} - Y_{j^-,:}\|_2 \leq C\left(\|X_{j,:} - X_{j^-,:}\|_2 + \sqrt{1 - s_{t,i}} + \gamma M\right),$$

where $C = L(1 + C')$ absorbs constants. □

