# OpenReview forum: "Fastcar: Cache Attentive Replay for Fast Auto-Regressive Video Generation on the Edge"
_ICLR.cc/2026/Conference — ICLR 2026 Poster_

### Official Review · Reviewer_jt8U · 2025-10-29

**Soundness:** 3
**Presentation:** 3
**Contribution:** 2
**Rating:** 6
**Confidence:** 3

**Summary:**

This paper proposes FASTCAR, an efficient framework for auto-regressive (AR) video generation, based on the insight that MLP modules are the main latency bottleneck and exhibit high temporal redundancy. FASTCAR implements a "cache attentive replay" strategy, using a Temporal Attention Score (TAS) derived from the preceding attention layer at no extra computational cost to determine whether to skip an MLP computation and reuse the cached output from the previous frame. Experimental results demonstrate that this method outperforms sparse attention approaches on VBench, achieving speedup with better generation quality and power efficiency on edge hardware.

**Strengths:**

1. This paper is well-written and easy to read.
2. The proposed method is simple, efficient and effective.
3. When combined with sliding window attention, FastCAR show the possibility to alleviate drifting caused by global information dropping.

**Weaknesses:**

1. The hardware design section is not connected to other parts and no experiment results prove its efficiency.
2. The claim that MLPs dominate latency is based on 8 frames short sequences which is relatively short to real-world using cases.
3. Lacks discussion of similar caching approaches, such as DeepCache in diffusion generation.
4. typo: "third" in line 459.

[1] [CVPR2024] DeepCache: Accelerating Diffusion Models for Free

**Questions:**

Though QK^T is computed in the attention layer, the result is temporary and on-chip in Flash Attention implementations. How do you calculate TAS efficiently when Flash Attention fusion is used? Does this require kernel modifications and what is the overhead?

---

> ### Author Response · Authors · 2025-11-22
> **Author rebuttal to reviewer jt8U for weakness 1**
>
> ### Rebuttal for Weakness 1
>
> Thank you for pointing this out.
> Our hardware design (DRS) is directly motivated by the algorithmic component (TAS).
> The TAS score provides a single-scalar gating signal for the computation of MLP modules, which introduces workload imbalance across different processing cores.
> The DRS is designed to balance this computational workload by constructing an on-chip computation-mapping table.
> When pre-compiled instructions are loaded and the replay mechanism is triggered, the prefetched instructions are forwarded to the DRS. The DRS consults the Index Register to decide whether instructions corresponding to replayed batches should be discarded or dispatched to the appropriate cores indicated by the Mapping Registers. Notably, the DRS completes this dispatch logic within only hundreds to thousands of cycles, which is negligible compared to the thousands of cycles required for the actual execution of MLP instructions.
> This shows that the hardware design is tightly coupled with the algorithm and introduces minimal overhead.
>
> Meanwhile, we provide more details for our hardware implementation target on the Xilinx Alveo U280 FPGA is provided in the table below.
> This card provides 8 GB of HBM (460 GB/s) and 32 GB of DDR4 (38 GB/s).
> Because the U280 is split into three discrete SLRs, long cross-die routes can easily become the critical paths that limit timing closure.
> To avoid this, we place independent accelerator cores in separate SLRs; each core is entirely contained within one die, eliminating inter-SLR logic.
> The HBM controller is physically located in SLR0 (the bottom SLR), so all memory-interface logic is also placed there.
> The entire design operates from a single 200 MHz global clock, and the detailed resource utilization is listed below.
>
> Our hardware implementation turns 69.74\% of the DSPs and 72.10\% BRAMs into performance.
> On-chip buffers built from BRAMs and URAMs enable fast data caching between HBM/DDR and computational logic.
> We implement 24 AXI channels, which can provide 72.10\% effective bandwidth utilization of HBM.
> By sufficient resource utilization, bandwidth utilization, and high clock frequency, our hardware design can achieve efficient deployment of autoregressive video generation models.
> Thanks again for the suggestion. We will provide more details in the revision.
>
> | Component              | LUT (Usage %)      | FF (Usage %)       | DSP (Usage %)          | BRAM (Usage %)             | URAM (Usage %)        |
> |-------------------------|:-----------------:|:------------------:|:----------------------:|:---------------------------:|:---------------------:|
> | SRAM                    | 126k (9.70%)      | 74k (2.84%)        | 0                      | 1194 (59.23%)              | 320 (33.33%)          |
> | DMA                     | 76k (5.81%)       | 48k (1.85%)        | 0                      | 171 (8.48%)                | 0                     |
> | MU                      | 454k (34.83%)     | 451k (17.29%)      | 6162 (68.28%)          | 0                          | 0                     |
> | VU                      | 104k (8.01%)      | 339k (13.12%)      | 128 (1.42%)            | 0                          | 0                     |
> | Control & Fetch & DRS   | 3k (0.27%)        | 4k (0.15%)         | 0                      | 13 (0.64%)                 | 0                     |
> | PCIe & DDR & HBM        | 49k (3.78%)       | 59k (2.27%)        | 3 (0.03%)              | 75.5 (3.75%)               | 0                     |
> | **Total**               | **813k (62.39%)** | **975k (37.39%)**  | **6293 (69.74%)**      | **1453.5 (72.10%)**        | **320 (33.33%)**      |

---

> ### Author Response · Authors · 2025-11-22
> **Author rebuttal to reviewer jt8U for weakness 2 and weakness 3**
>
> ### Rebuttal for Weakness 2
>
> The choice of 8-frame input sequences follows the default setting of the VILA-U model and its official open-source inference pipeline.
> Our method is not limited to this setting and can support more frames, i.e., longer token-length generation during decoding.
> As shown in Figure 1, the MLP still dominates the latency even when the sequence length increases to 8K tokens (i.e., 32 frames), indicating that our approach remains effective for longer sequences.
>
> Furthermore, when longer sequences lead to higher attention costs, sparse-attention techniques can be introduced to accelerate the attention module. Our method is orthogonal to such approaches, as demonstrated by the combined results in Table 2 and Table 7. Note that our method alleviates the drifting brought by the sparse attention methods as shown in Figure 6.
>
> Therefore, we believe our method remains effective and relevant for longer videos and real-world settings.
>
> ### Rebuttal for Weakness 3
>
> Thanks for the suggestion. We highlight that our method is foundamentally different from previous caching approaches like DeepCache for diffusion models. We specify the differences below and will add more discussions in the revision.
>
> (i) The caching are applied to different contents. Our method replays MLP outputs for generating new tokens of new video frames during decoding. In contrast, DeepCache is adopted to refine the same latent frame.
>
> (ii) The strategy of when to cache and replay is significantly different. Our method adopts attention-based TAS as a metric  to determine whether to replay with theoretical support.  Different from our runtime TAS computated during inference, the caching decision in DeepCache is heuristic, by manually choosing the skip branches and determining the caching intervals before actual inference.
>
> (iii) As a result, the caching stragety in DeepCache is fixed for generating different images (such as perform caching every 3 steps at fixed locations). However, our replay is dynamically detemined by the TAS computation during runtime, with various replay ratios across different layers for different video generations, as shown in Figure 5.
>
> (iv) Our method replays the post-attention representation (i.e., the input of MLP modules), which is different from DeepCache to cache block outputs  in the U-Net.
>
> (v) Our theoretic discussion in Theorem 4.6 formally   justifies the use of thresholds on TAS to dynamically skip MLP computations during decoding with guarantees on small error bound.
> DeepCache does not provide a theoretic analysis, and manually skips the blocks every a few steps under a heuristic fixed stragegy, without a similarity metric or threshold like ours.
>
> Thanks for the comments. To the best of our knowledge, no prior work explores reusing hidden states or skipping MLP computation in AR video generation.
> We will include the discussion with previous cahce based methods in our revision.

---

> ### Author Response · Authors · 2025-11-22
> **Author rebuttal to reviewer jt8U for question 1**
>
> ### Rebuttal for Question 1
>
> Our hardware design is efficient.  The DRS  decides whether instructions corresponding to replayed batches should be discarded or dispatched.
> The DRS completes this logic within only hundreds to thousands of cycles, which is negligible compared to the thousands of cycles required for the actual execution of MLP instructions.
>
> Even if  the Flash Attention is imported, our method still leads to superior efficiency, as TAS does not require access to the full attention matrix or the fused $Q K^T$ output.
> As illustrated in Eq. (7), only the current query vector (a single token during decoding), and the key vector at the same spatial location from the previous frame (also a single token) are used for the TAS score computation, which incurs negligible overhead and does not require any modification to the Flash Attention kernel.
>
> Thus,  in systems using Flash Attention fusion, TAS computation remains extremely lightweight and fully decoupled from the fused attention kernel.

---

### Official Review · Reviewer_wS7K · 2025-10-29

**Soundness:** 2
**Presentation:** 3
**Contribution:** 2
**Rating:** 4
**Confidence:** 2

**Summary:**

This paper proposes FastCar, a method to speed up autoregressive video generation by avoiding redundant MLP computation across adjacent frames. The authors profile decoding latency and argue that the MLP/FFN block, not attention, is the main bottleneck. They observe that many tokens barely change between consecutive frames, so instead of recomputing the MLP output every time, FastCar “replays” cached MLP outputs from the previous frame for stable tokens. A Temporal Attention Score, derived from cross-frame attention, is used to decide which tokens can be safely replayed, and the paper provides a theoretical argument that high TAS implies similar MLP outputs.

The paper further presents an FPGA-oriented runtime scheduler (DRS) that dynamically assigns only the “needs recompute” tokens to compute cores to keep utilization high. Experiments claim >2× speedup over dense decoding and better quality / less drift than sparse-attention baselines. The high-level idea of reusing features for unchanged regions is conceptually simple and related to prior token-reuse / feature-caching work, but the authors position their novelty in applying it to autoregressive video decoding, tying the gating decision to TAS, and demonstrating a hardware-aware implementation.

**Strengths:**

The paper performs actual latency profiling and shows that, in autoregressive video decoding, the main bottleneck is the MLP/FFN block rather than attention. This is valuable because most prior acceleration work focuses on attention; here the motivation is concrete and data-driven.

The core mechanism of FastCar — reusing the previous frame’s MLP outputs for tokens that barely change, instead of recomputing them every step — is straightforward and can be applied at inference time without retraining. The per-token, per-layer replay design makes it easy to imagine plugging this into existing AR video generators as a decoding-time optimization.

The method uses a Temporal Attention Score (TAS), derived from cross-frame attention, to decide which tokens can safely skip recomputation. The paper also provides a theoretical argument that high TAS implies similar MLP outputs. This lifts the approach above a pure heuristic and makes the selective replay decision more interpretable.

Beyond the algorithmic idea, the paper presents a hardware-oriented runtime scheduler (DRS) on FPGA that dynamically dispatches only the “needs recompute” tokens to available compute cores, improving utilization and energy efficiency. This gives the work a full-stack feel rather than just a conceptual trick.

The experiments report over 2× decoding speedup while maintaining generation quality and reducing long-horizon drift compared to sparse-attention baselines. This suggests the approach could be especially useful for long-duration or higher-resolution AR video generation.

**Weaknesses:**

1. The core idea — detect tokens/regions that barely change over time and skip recomputing them by reusing cached features — is not fundamentally new. Similar per-token reuse / feature caching / dynamic skipping ideas have already appeared in video and diffusion generation acceleration. The paper mainly adapts this intuition to autoregressive video decoding, adds a specific gating signal (TAS), and wraps it with an FPGA story. This feels more like an incremental systemization than a genuinely new algorithmic principle.

2. And as written, all results (latency profiling, TAS analysis, replay ablations, quality vs. drift comparisons, and the FPGA/DRS speedup and energy numbers) appear to be reported on a single autoregressive video generation model (vila-u-7b-256)? Could you explain why the evaluation is limited to this one model? In particular: (1) is FastCar tied to architectural details of that specific model (e.g., tokenizer, decoder layout, MLP shape), or should it in principle apply to other AR video generators? (2) did you attempt to run the method on any other AR video backbones or model sizes, and if so, what prevented you from reporting those results? Right now it is difficult to judge how general the proposed approach is versus a case study on one model.

If I missed any details, please let me know during the rebuttal period.

**Questions:**

See Weaknesses

---

> ### Author Response · Authors · 2025-11-22
> **Author rebuttal to reviewer wS7K for weakness 1**
>
> ### Rebuttal for Weakness 1
>
> Thanks for the comment. We make significant contributions to identify  and formalize temporal redundancy at the token level inside auto-regressive video generation. To the best of our knowledge, no prior work explores reusing hidden states or skipping MLP computation in AR video generation.
> Though the high level intuition of “reusing similar features” has appeared in several prior caching approaches  such as MagCache [1] and AdaCache [2] for video diffusion models, our work is fundamentally different from these works as detailed below.
>
> (i) The caching are applied to skip different temporal  contents. Our method uses cache to skip certain positions for adjacent video frames during decoding. In contrast, MagCache or AdaCache is adopted to skip certain denoising steps.  We explore the temporal redundancy in consecutive frames, different from other diffusion-based caching for redundancy in denoising steps.
>
> (ii) Our theoretic discussion in Theorem 4.6 formally   justifies the use of thresholds on TAS to dynamically skip MLP computations during decoding with guarantees on small error bound.
> Other methods such as MagCache and AdaCache mainly use heuristic metrics without a detailed theoretic  analysis.
>
> (iii) Our method replays the post-attention representation (i.e., the input of MLP modules in the decoding), which is different from other works to cache block outputs  in the diffusion parts with U-Net or Diffusioin Transformer.
>
> (iv) Our hardware design (DRS) is directly motivated by our proposed  algorithmic component (TAS).
> TAS provides a single-scalar gating signal for the computation of MLP modules, which introduces workload imbalance across different processing cores. To address this, the corresponding hardware accelerator, DRS, is designed to balance this computational workload by constructing an on-chip computation-mapping table. It allows selective computation reuse through a lightweight, single-cycle gating mechanism.
> Other works do not consider this workload imbalance issue during caching.
>
> Our algorithmic design with theoretic support is significantly different from prior works. Meanwhile,  our hardware design improves resource utilization and accelerates inference on edge devices, demonstrating that our algorithmic approach is not simply conceptual but fully deployable in real systems. It is tightly coupled with the algorithm and introduces minimal overhead.  This combination has not been explored in prior acceleration efforts and is essential for achieving real-time AR video generation on resource-constrained platforms.
> Thanks for the comments.  We will include more  discussions with previous works in our revision.
>
> -----
>
> [1] MagCache: Fast Video Generation with Magnitude-Aware Cache
>
> [2] Adaptive Caching for Faster Video Generation with Diffusion Transformers

---

> ### Author Response · Authors · 2025-11-22
> **Author rebuttal to reviewer wS7K for weakness 2**
>
> ### Rebuttal for Weakness 2
>
> **Broad Applicability.**
>
> Many thanks for the question. FastCar is designed to be broadly applicable to  token-level AR video generation models.
> Our method relies only on two architectural properties that are widely shared across modern AR Transformers:
>
> (1) *Standard Transformer block architecture*:
> TAS is defined purely based on the standard attention mechanism: it measures the temporal consistency between adjacent frames using the attention map, and the resulting score determines whether the subsequent MLP computation can be skipped. Therefore, TAS does not depend on hidden size, MLP expansion ratio, or any VILA-U-specific setting. Our method works when there is conventional multi-head attention followed by an MLP, which is common to nearly all AR Transformer decoders.
>
> (2) *Token-level auto-regressive decoding*:
> FastCar operates at the token level and does not assume any particular tokenizer design. Whether the visual tokens are produced by VQ-VAE or other emerging tokenizers, the replay mechanism only requires that the model predicts the next token in the auto-regressive format. The temporal redundancy captured by TAS arises from the video content and the model’s attention behavior, not from the tokenization scheme.
>
> **The Only Available Model VILA-U.**
>
> Our choice of VILA-U is primarily driven by the current ecosystem of open-source token-level Auto-Regressive (AR) video generation models: VILA-U is the only publicly available model that follows a standard Transformer architecture with token-wise AR decoding.
> Other AR video generations like NOVA [1] adopts frame-by-frame prediction, where an entire frame is treated as the atomic prediction unit.
> This frame-level decomposition produces a much coarser granularity than token-level decoding, making it unsuitable for our edge-level deployment (i.e., FPGA in our work), where fine-grained token-level decoding is adopted.
> Thus, for the generalization to the token-level AR video generation architecture, our framework is directly applicable.
>
> **Why not Image AR Models.**
>
> As for text or image AR models, those models do not contain adjacent temporal frames, and thus lack the cross-frame redundancy that our method exploits. For these domains with one single frame, our framework to reuse results from previous frames is not directly applicable.
>
> **MLP-Focused Plug-and-Play Enhancement.**
>
> For autoregressive video models that do include diffusion progress, our method can still be used to accelerate the token-level decoding phase, since FastCar is orthogonal to the diffusion acceleration techniques.
> Nevertheless, because our goal is to support deployment on edge devices with limited resources, we focus on VILA-U—whose purely autoregressive structure simplifies hardware implementation and allows us to prototype an efficient FPGA execution pipeline.

---

> ### Author Response · Authors · 2025-11-25
>
> Dear Reviewer wS7K,
>
> We hope you are doing well. We would like to kindly check in regarding the review of our rebuttal. We sincerely appreciate the time and effort you have already put into evaluating our submission, and we understand that reviewing can be time-consuming. If there are any remaining questions or clarifications needed on our side, we would be more than happy to provide additional details.
>
> Thank you again for your time and consideration.
>
> Best,
> Authors

---

### Official Review · Reviewer_RrDJ · 2025-10-31

**Soundness:** 3
**Presentation:** 3
**Contribution:** 3
**Rating:** 6
**Confidence:** 4

**Summary:**

This paper proposes FastCar, a system-level framework to accelerate auto-regressive (AR) video generation on edge devices by exploiting temporal redundancy in MLP outputs across adjacent video frames. The core insight is that MLP modules, not attention, dominate decoding latency in AR video models like VILA-U, and their outputs exhibit high similarity between consecutive frames. To leverage this, the authors introduce the Temporal Attention Score (TAS), a lightweight metric derived from attention logits, to decide when to replay (i.e., reuse) cached MLP outputs from the previous frame instead of recomputing them. They further design a custom FPGA-based hardware accelerator with Dynamic Resource Scheduling (DRS) that adapts computation allocation in real time based on TAS-driven replay decisions. Experiments show that FastCar achieves >2.1x decoding speedup, ~45% FLOPs reduction, and higher energy efficiency than sparse attention baselines, while preserving video quality and even mitigating temporal drifting when combined with sparse attention.

**Strengths:**

1. Strong empirical motivation: The paper provides compelling profiling evidence that MLPs, not attention, are the bottleneck in AR video decoding, which justifies shifting optimization focus away from KV-caching or sparse attention (common in LLMs) toward MLP replay.
Novel and well-motivated algorithmic component: The Temporal Attention Score (TAS) is a simple yet effective proxy for temporal similarity that incurs zero extra compute (as it reuses existing attention logits). The theoretical analysis (Theorems 4.4–4.6) formally links TAS to MLP output similarity, lending credibility to the replay strategy.
2. Hardware-software co-design: The integration of TAS with a custom FPGA accelerator featuring DRS is a notable strength. The DRS mechanism intelligently balances workload across cores in response to dynamic replay patterns, which is essential for real-world deployment.
3. Comprehensive and reproducible evaluation: The authors evaluate across multiple replay ratios, report detailed metrics (PSNR, SSIM, LPIPS, VBench, latency, power, FLOPs), compare against StreamingLLM (a relevant sparse attention baseline), and demonstrate complementarity with sparse attention.
4. Practical relevance: The work directly addresses a critical bottleneck in deploying AR video models on edge devices, making it highly relevant to both systems and generative AI communities.

**Weaknesses:**

1. Threshold selection is manual: The replay threshold is tuned empirically. The paper shows robustness across thresholds (Fig. 4), but does not propose an adaptive or learned thresholding strategy, which could improve usability in dynamic real-world scenarios.
2. Limited evaluation and comparison: Evaluation limited to VILA-U, comparison limited to StreamingLLM. Yes, VILA-U is the only open-source AR video generation without diffusion model, but it may indicate that the community and business are not interested in such types of models.

**Questions:**

Generality beyond VILA-U: Is FastCar applicable to other AR video models that may have different MLP/attention ratios or tokenization schemes? What architectural properties are required for TAS to be predictive?

---

> ### Author Response · Authors · 2025-11-22
> **Author rebuttal to reviewer RrDJ for weakness 1**
>
> ### Rebuttal for Weakness 1
>
> Thanks for the suggestions. We adopt a consistent threshold since we find  that it is  empirically effective with theoretical support, and hardware-efficient  in our framework.  We will keep  investigating this direction with more sophisticated, adaptive thresholding mechanisms   for further breakthroughs.
>
> As shown in Figure 4, we tried a simple strategy with layer-wise varying (i.e., inconsistent) thresholds across all layers. Figure 4 demonstrates that a single global threshold is more effective with   better performance  in terms of generation quality and stability, than inconsistent thresholds.  This empirical observation supports the theoretical insight in Theorem 4.4 and Remark 4.7 (Section 4.4), which demonstrate through detailed analysis that TAS provides a reliable replay trigger. Specifically, the derived error bounds do not depend on model depth, indicating that TAS is inherently layer-wise insensitive. This theoretical property explains why a consistent threshold across layers yields better performance (as shown in Figure 4), and suggests that consistent threshold is both effective and easier to deploy across diverse architectures.
>
> Furthermore, although we adopt a consistent threshold, the replay ratios of different layers follows non-uniform distributions, demonstrating adaptive adjustment capability in the model itself. Speficially, Figure 5 shows the replay-ratio distribution across layers under different threshold settings. We identify that, even with a fixed threshold, the replay behavior across layers is already highly non-uniform: shallow and deep layers tend to be replayed more frequently than intermediate layers. This naturally emerging pattern indicates that the TAS signal itself induces an implicit form of adaptivity without requiring an explicitly adaptive or per-layer thresholding mechanism.
>
> Moreover, from a hardware perspective, a global consistent threshold is more hardware-efficient than adaptive thresholds which may substantially complicate the accelerator design. Specifically, a dynamic threshold requires runtime collection of TAS statistics and additional control logic to update the value, and these updates introduce synchronization barriers and potential pipeline stalls. Such operations incur non-negligible on-chip memory and control overhead, and break the fully pipelined  deterministic execution path used in our FPGA implementation.
> In contrast, a fixed global threshold enables a lightweight, single-cycle comparator per token and ensures stable latency and predictable resource usage—properties that are essential for real-time edge deployment.
>
> Based on the above, we adopt a consistent threshold in our paper. Thanks for the comment, we will keep investigating this direction with adaptive thresholds for furthre breakthroughs.

---

> ### Author Response · Authors · 2025-11-22
> **Author rebuttal to reviewer RrDJ for weakness 2**
>
> ### Rebuttal for Weakness 2
>
> Many thanks for the comment. We agree with the reviewer on the limited availability of open-source AR video generation models without diffusion.
> Despite the scarcity, we believe that AR video generation models remains highly important, and will enjoy increasingly popularity in the future.
> Autoregressive video generators built on discrete visual tokens provide a unified generative interface across video, image, and language modalities. Such unified formulation is particularly valuable for interactive and multimodal systems. As a result, token-based unified generation models are increasingly popular in both industry and the research community, such as [1] [2] [3].
>
> Furthermore, our method only  optimizes the MLP part without changing other model parts, and can be integrated with other methods which optimize other parts in the model. We demonstrate an example which combines our method and sparse attention in Table 2. It shows that our method can sginificantly enhance the sparse attention method and  alleviate its drifting issues. Our proposed framework is orthogonal and  complimentary  to different model-level improvements, with broad general applicapability.
>
> We specify the rational for the comparison with StreamingLLM below.
> (i) MLP computations in the decoding phase, rather than attention, dominate the inference latency, as shown in Figure 1 and our motivation in Section 3. Inspired by this, we focus on optimizing the MLP part in decoding.
> (ii)  Since  StreamingLLM directly optimizes decoding-time efficiency, it serves as the most relevant baseline to ours which also focuses on decoding efficiency.
> (iii) Most other block-wise sparse attention methods focus primarily on the prefill phase. It may not be fair to compare them with our decoding-focused method.
> Thus, we compare with StreamingLLM without including a broader range of sparse-attention approaches.
>
> Furthermore, by combining our method and sparse attention, a better generation performance can be achieved. As shown in Figure 6, when applying both techniques, our method alleviates the drifting effects introduced by sparse attention. This further supports the effectiveness of our framework and shows that it can be integrated with sparse attention techniques in AR video generation models.
>
>
> ----
>
> [1] Generative Multimodal Models are In-Context Learners
>
> [2] UniToken: Harmonizing Multimodal Understanding and Generation through Unified Visual Encoding
>
> [3] MMaDA: Multimodal Large Diffusion Language Models

---

> ### Author Response · Authors · 2025-11-22
> **Author rebuttal to reviewer RrDJ for question 1**
>
> ### Rebuttal for Question 1
>
> Thank you for the question. FastCar is not tied to VILA-U and is designed to be broadly applicable to other token-level AR video generation models.
> Our method relies only on two architectural properties that are widely shared across modern AR Transformers:
>
> (1) *Standard Transformer block architecture*:
> TAS is defined purely based on the standard attention mechanism: it measures the temporal consistency between adjacent frames using the attention map, and the resulting score determines whether the subsequent MLP computation can be skipped. Therefore, TAS does not depend on hidden size, MLP expansion ratio, or any VILA-U-specific setting. Our method works when there is conventional multi-head attention followed by an MLP, which is common to nearly all AR Transformer decoders.
>
> (2) *Token-level auto-regressive decoding*:
> FastCar operates at the token level and does not assume any particular tokenizer design. Whether the visual tokens are produced by VQ-VAE or other emerging tokenizers, the replay mechanism only requires that the model predicts the next token in the auto-regressive format. The temporal redundancy captured by TAS arises from the video content and the model’s attention behavior, not from the tokenization scheme.
>
> For models with different MLP/attention FLOP ratios, FastCar remains applicable.
> If the MLP dominates the decoding cost, our method provides substantial acceleration. If attention becomes the dominant component, FastCar still removes redundant MLP computation and serves as an orthogonal optimization that can be combined with existing attention-acceleration techniques.

---

### Official Review · Reviewer_XRWX · 2025-11-01

**Soundness:** 2
**Presentation:** 3
**Contribution:** 3
**Rating:** 6
**Confidence:** 3

**Summary:**

This paper aims to address the high computational cost of auto-regressive (AR) video generation, particularly for deployment on edge devices. The authors argue that in AR video generation, the MLP modules—not the attention modules—dominate the inference latency during the decoding phase. They also observe a high degree of temporal redundancy in the outputs of these MLP modules for adjacent frames.

Based on these findings, they propose FastCar, a framework that accelerates decoding by conditionally reusing cached MLP outputs from the previous frame. The decision to reuse or recompute is governed by a novel metric called the Temporal Attention Score (TAS), which measures the similarity between a token's query and the key of its corresponding token in the prior frame.

Experimental results on the VILA-U model show that FastCar significantly outperforms sparse attention methods, achieving over a 2.1x speedup when combined with sparse attention, while better preserving generation quality and improving energy efficiency.

**Strengths:**

1. The paper is well-written. The proposed method is well-illustrated and easy to follow.

2. **The proposed method is simple and efficient:** Using the Temporal Attention Score (TAS)—a metric that is effectively "free" as it is derived from pre-existing attention calculations—to guide the caching strategy is a very clever design choice. This avoids the overhead that often plagues other dynamic execution methods.

3. **Comprehensive empirical evaluation:** The paper presents a robust set of experiments with convincing results. The method demonstrates significant improvements in latency, throughput, and power efficiency. The comparison against sparse attention (StreamingLLM) effectively highlights the superiority of the proposed approach for this problem domain. Furthermore, showing that FastCar can be combined with sparse attention to mitigate its weaknesses (like "drifting") and achieve even greater speedups is a powerful demonstration of its complementary nature.

4. **Practical hardware co-design and validation:** This work goes beyond a purely algorithmic proposal by designing, implementing, and evaluating a hardware accelerator on an FPGA. The development of Dynamic Resource Scheduling (DRS) to handle the dynamic workloads shows a deep understanding of the practical challenges involved in deploying such models. This end-to-end, software-hardware co-design approach greatly increases the credibility and practical value of the research.

**Weaknesses:**

1. **Limited architectural generalization:** The experiments and analysis are conducted exclusively on the VILA-U model. While the authors correctly note the lack of other open-source AR video models, this raises a question about the generality of the core insight. The MLP-bottleneck observation may be specific to this particular architecture's configuration (e.g., hidden size, MLP expansion factor). The paper would be stronger if it included analysis on other AR models (even from image or language domains) to hypothesize about the broader applicability of the findings.

2. **Static thresholding mechanism:** The replay decision relies on a single, manually-tuned, and globally consistent threshold (`τ`). While effective, this is a relatively simple criterion. A more sophisticated, adaptive thresholding mechanism could potentially unlock a better trade-off between performance and quality. The ablation on "inconsistent thresholds" is a good first step, but since the thresholds were manually set, it doesn't fully explore the potential of adaptive strategies.

**Questions:**

1. The Temporal Attention Score (TAS) is calculated by averaging scores across all attention heads. Did you investigate the behavior of individual heads? Is it possible that some heads are more specialized in tracking temporal correspondence and could serve as better indicators for the replay decision than a simple average?

---

> ### Author Response · Authors · 2025-11-22
> **Author rebuttal to reviewer XRWX for weakness 1**
>
> ### Rebuttal for Weakness1
>
> **Broad Applicability.**
>
> Thank you for the question. FastCar is designed to be broadly applicable to  token-level Auto-Regressive (AR) video generation models.
> Our method relies only on two architectural properties that are widely shared across modern AR Transformers:
>
> (1) *Standard Transformer block architecture*:
> TAS is defined purely based on the standard attention mechanism: it measures the temporal consistency between adjacent frames using the attention map, and the resulting score determines whether the subsequent MLP computation can be skipped. Therefore, TAS does not depend on hidden size, MLP expansion ratio, or any VILA-U-specific setting. Our method works when there is conventional multi-head attention followed by an MLP, which is common to nearly all AR Transformer decoders.
>
> (2) *Token-level auto-regressive decoding*:
> FastCar operates at the token level and does not assume any particular tokenizer design. Whether the visual tokens are produced by VQ-VAE or other emerging tokenizers, the replay mechanism only requires that the model predicts the next token in the auto-regressive format. The temporal redundancy captured by TAS arises from the video content and the model’s attention behavior, not from the tokenization scheme.
>
> **The only Available Model VILA-U.**
>
> Our choice of VILA-U is primarily driven by the current ecosystem of open-source token-level AR video generation models: VILA-U is the only publicly available model that follows a standard Transformer architecture with token-wise AR decoding.
> Other AR video generations like NOVA [1] adopts frame-by-frame prediction, where an entire frame is treated as the atomic prediction unit.
> This frame-level decomposition produces a much coarser granularity than token-level decoding, making it unsuitable for our edge-level deployment (i.e., FPGA in our work), where fine-grained token-level decoding is adopted.
> Thus, for the generalization to the token-level AR video generation architecture, our framework is directly applicable.
>
> **Why not Image AR models.**
>
> As for text or image AR models, those models do not contain adjacent temporal frames, and thus lack the cross-frame redundancy that our method exploits. For these domains, our framework is not directly applicable.
>
> **MLP-Focused Plug-and-Play Enhancement.**
>
> We focus on the MLP acceleration during decoding, which can be seamlessly combined with other sparse attention methods.
> For models with different MLP/attention FLOP ratios, FastCar remains applicable.
> If the MLP dominates the decoding cost, our method provides substantial acceleration. If attention becomes the dominant component, FastCar still removes redundant MLP computation and serves as an orthogonal optimization that can be combined with existing attention-acceleration techniques.  As shown in Figure 6, when combined with sparse attention, our method alleviates the drifting effects introduced by sparse attention, demonstrating our plug-and-play enhancement without any additional model training.
>
> We will clarify this constraint more explicitly in the revised version.
>
> ---
>
> [1] Autoregressive Video Generation without Vector Quantization

---

> ### Author Response · Authors · 2025-11-22
> **Author rebuttal to reviewer XRWX for weakness 2**
>
> ### Rebuttal for Weakness2
>
> Thanks for the suggestions. We adopt a consistent threshold since we find  that it is  empirically effective with theoretical support, and hardware-efficient  in our framework.  We will keep  investigating this direction with more sophisticated, adaptive thresholding mechanisms   for furthre breakthroughs.
>
> The theoretical insights in Theorem 4.4 and Remark 4.7 (Section 4.4)  demonstrates through detailed analysis that TAS provides a reliable replay trigger. Specifically, the derived error bounds do not depend on model depth, indicating that TAS is inherently layer-wise insensitive. This theoretical property explains why a consistent threshold across layers yields better performance, and suggests that consistent thresholding is both effective and easier to deploy across diverse architectures. The theoretical analysis is further supported by our experimental observations in  Figure 4 with layer-wise varying (i.e., inconsistent) thresholds across all layers.
>
> In addition, Figure 5 provides the replay ratio distribution across layers under different thresholds. Interestingly, even with a fixed threshold, the replay behavior across layers is already highly non-uniform. We identify that the shallow and deep layers replayed more frequently than the intermediate layers. This naturally emerging pattern indicates that the TAS signal itself induces an implicit form of adaptivity, even without explicitly designing a per-layer adaptive threshold.
>
> Moreover, from a hardware perspective, a global consistent threshold is more hardware-efficient than adaptive thresholds which may substantially complicate the accelerator design. Specifically, a dynamic threshold requires runtime collection of TAS statistics and additional control logic to update the threshold.
> These updates also introduce pipeline stalls to ensure correct synchronization across layers and tokens.
> Such operations incur non-negligible on-chip memory overhead and break the fully pipelined and deterministic control path used in our FPGA implementation. % Written by Weize
> In contrast, a fixed global threshold allows a lightweight, single-cycle comparator per token and ensures stable latency and predictable resource usage, which are essential for real-time edge deployment.

---

> ### Author Response · Authors · 2025-11-22
> **Author rebuttal to reviewer XRWX for question 1**
>
> ### Rebuttal for Question 1
>
> Thank you for the insightful question.
>
> Exploring head-level redundancy was indeed one of the first things we investigated at the early stage of this work. However, we ultimately decided not to use head-wise TAS for the replay decision for the following reasons.
>
> First, head-level redundancy reflects redundancy inside the attention module, whereas Figure 1 shows that attention is not the main latency bottleneck during decoding. Thus, our acceleration primarily targets the MLP modules, where most FLOPs are concentrated.
>
> Second, and more importantly, head-level signals cannot be effectively propagated to the MLP stage for practical attention replay. After multi-head attention, all head outputs are projected and fused through the output projection matrix (i.e., $W_o$), producing a single hidden representation. By the time the representation enters the MLP block, the contributions of individual heads are linearly mixed and normalized. Thus, it is fundamentally difficult to make per-head replay for the reductioin of MLP computation.
>
> Therefore, when we focus on reducing redundancy in the MLP modules during decoding, we abandoned the fine-grained head-level design and chose the average score. Additionally, using the average score greatly simplifies the FPGA implementation, since it requires only a single scalar comparison per token rather than per-head statistics.

---

### Author Response · Authors · 2025-12-03

We sincerely thank the Area Chair and all reviewers for their  detailed feedback  and positive assessment of our work. We deeply appreciate the constructive suggestions, and we have carefully addressed every concern with both conceptual clarifications and new supporting evidence.

In response to the comments, we have provided corresponding answers and refinements throughout the rebuttal. The key improvements and clarifications include:

**Broader applicability of FastCar**: We clarified why FastCar is architecture-agnostic and applicable to general token-level transformer-based AR video generation models, and the VILA-U is the only open-source model that matches this setting.

**Justification for consistent thresholding**: We added theoretical insights, empirical evidence, and hardware motivations explaining why a global threshold is effective, stable, and hardware-friendly.

**Differentiation from prior caching work**: We clarified the fundamental differences between FastCar and existing caching/feature-reuse techniques in diffusion models and video generation.

**Hardware integration**: We strengthened the connection between TAS and the DRS hardware design, and explained the negligible overhead of the scheduling pipeline.

**Longer-sequence relevance**: We clarified that MLP dominance persists even at longer sequence lengths and showed that FastCar remains useful in real-world long video scenarios.

We believe these explanations significantly strengthen our submission and further highlight the novelty, generality, and practical value of FastCar.

---

### Meta-Review · Area_Chair_h7cq · 2026-01-09

**Summary:**

This paper proposes FastCar, a system-level framework to accelerate auto-regressive (AR) video generation on edge devices by exploiting temporal redundancy in MLP outputs across adjacent video frames. The paper received scores of 4666. All the reviewers appreciated the strong empirical motivation, hardware-software co-design, and the comprehensive evaluation. A common concern, however, was that all experiments were conducted on a single model, VILA-U, raising questions about the generalizability of the approach to other AR video generation models. The authors argued that VILA-U is the only open-source AR video generation model for this setting. Overall, I find the rebuttal convincing and recommend acceptance.

**Reviewer Concerns:**

Concerns adequately addressed:

1. Justification for consistent thresholding: The authors added theoretical insights, empirical evidence, and hardware motivations.

2. Differentiation from prior caching work: The authors made corresponding clarifications during rebuttal.

3. Hardware integration: The connection between TAS and the DRS hardware design was strengthened.

4. Longer-sequence relevance: The authors clarified that MLP dominance persists even at longer sequence lengths and showed that FastCar remains useful in real-world long video scenarios.


Concerns insufficiently addressed:

All the reviewers showed the common concern that the experiments and analysis are conducted exclusively on the VILA-U model. The authors argued that the proposed method itself is general, and VILA-U is the only open-source model that matches this setting. This argument makes sense to the AC, though not entirely satisfying.

**Reviewer Scores:**

Overall, I think the reviewer who assigned a score of 4 has reasonable grounds to increase it to 6 after considering the rebuttal. In summary, the AC finds the rebuttal convincing and supports acceptance.

---

### Decision · Program_Chairs · 2026-01-26

Accept (Poster)